



# Enhancing Urban Pluvial Flood Modelling through Graph Reconstruction of Incomplete Sewer Networks

Ruidong Li[1], Jiapei Liu[1], Ting Sun[2], Shao Jian[3], Fuqiang Tian[1], and Guangheng Ni[1]

[1]State Key Laboratory of Hydro-science and Engineering, Department of Hydraulic Engineering, Tsinghua University, Beijing 100084, China
[2]Institute for Risk and Disaster Reduction, University College London, London WC1E 6BT, UK
[3]Yinchuan Meteorological Bureau, Yinchuan 750002, China

**Correspondence:** Ruidong Li (lyy0744@mail.tsinghua.edu.cn) and Ting Sun (ting.sun@ucl.ac.uk)

**Abstract.**

This work presents an efficient graph reconstruction-based approach for generating physical sewer models from incomplete information, addressing the challenge of representing sewer drainage effect in urban pluvial flood simulation. The approach utilizes graph-based topological analysis and hydraulic design constraints to derive gravitational flow directions and nodal in-
vert elevations in decentralized sewer networks with multiple outfalls. By incorporating linearized programming formulation to solve reconstruction problems, this approach can achieve high computational efficiency, enabling application to city-scale sewer networks with thousands of nodes and links. Tested in Yinchuan, China, the approach integrates with a 1D/2D coupled hydrologic-hydrodynamic model and accurately reproduces maximum inundation depths ($R^2 = 0.95$) when the complete network layout and regulated facilities are available. Simplifications, such as adopting road-based layouts and omitting regulation
facilities, can degrade simulation performance for extreme rainfall events compared to calibrated equifinal methods. However, design rainfall analysis demonstrates that the physical reconstruction approach can reliably outperform equifinal methods, achieving reduced variation and higher accuracy in simulating inundation areas. However, proper configuration of regulated facilities and network connectivity remains crucial, particularly for simulating local inundation during extreme rainfall. Thus, it is recommended to integrate the proposed algorithm with targeted field investigations to further improve urban pluvial flood
simulation performance in data-scarce regions.

## 1 Introduction

Global climate changes and intense human activities significantly alter climatic backgrounds and surface environments, leading to an increase in the frequency and magnitude of urban pluvial floods induced by excessive rainfalls and outdated drainage systems designed for past situations (Rosenzweig et al., 2018). As representative man-made infrastructure, sewer networks
can effectively mitigate pluvial flood risks by rapidly transporting rainwater to downstream sites such as river channels and detention ponds (Wang et al., 2022). However, its performance can be inherently insufficient and gradually deteriorate due to improper design and poor maintenance (Tran et al., 2024), further causing overflow and exacerbating surface floods during



extreme rainfall (Schmitt and Scheid, 2020). Thus, an accurate representation of the sewer drainage effect is essential for urban pluvial flood simulation and risk attribution (Luan et al., 2024).

However, detailed sewer network information may not always be readily available and its implementation in hydraulic models requires laborious work with high uncertainty (Montalvo et al., 2024), yielding multiple simplified approaches to approximate the sewer drainage effect. Most approaches assume a term of sewer-induced mass loss during the rainfall-runoff process in specified regions, such as road areas and inlet areas (Li et al., 2020; Xing et al., 2021, 2019). However, they ignore the spatiotemporal distribution of hydraulic head at rainwater inlets and therefore fail to simulate the phenomenon of node

overflow due to overload. Thus, Montalvo et al. (2024) proposes a physics-based approach which approximates the effects of the actual sewer network by designing a corresponding virtual one with open-access data and local regulations (Reyes-Silva et al., 2023). This approach can establish a sewer hydraulic model and allow for the simulation of the bidirectional interaction between the ground surface and the sewer system and therefore gives a more comprehensive understanding of urban pluvial flood process.

The virtual network generation (VNG) proposed in this study closely follows the steps of sewer network design (SND), encompassing two main components for both water distribution and urban drainage networks (Duque et al., 2016):

- **Layout Definition:** This step establishes geometric connections and hydraulic directions between nodes in the sewer network. Various algorithms have been developed, including graph theory-based modelling (Bakhshipour et al., 2019; Kim et al., 2021), surface elevation-driven delineation (Blumensaat et al., 2012; Duque et al., 2022), and street network-

guided configuration (Chegini and Li, 2022; Reyes-Silva et al., 2023). Most existing work simplifies the target system with a centralized layout where a single outfall node is present (Hesarkazzazi et al., 2022), allowing the use of tree-based algorithms such as the minimum spanning tree (Reyes-Silva et al., 2023) and the Steiner minimal tree (Machine Hsie and Huang, 2019). However, this simplification ignores the rich topological structures of decentralized networks with potential cycles and forests (Bakhshipour et al., 2019). Additionally, most approaches assume that the direction of gravity

flow follows the surface slope, which is not always accurate, especially in flat areas or with inaccurate elevation data (Hesarkazzazi et al., 2022; Dunton and Gardoni, 2024).

- **Hydraulic Design:** This step determines the sizes and slopes of pipes, as well as the necessary configuration of rainwater treatment facilities where the goal is to fulfill the design discharge requirement estimated by the rational method (Wang and Wang, 2018; Reyes-Silva et al., 2023). Most approaches solve the corresponding problem using mathematical pro-

gramming, including linear programming (Swamee and Sharma, 2013), mixed-integer linear programming (Safavi and Geranmehr, 2017), nonlinear programming (Li and Matthew, 1990), and multi-objective programming (Bakhshipour et al., 2021). However, significant computational efforts are required to obtain global optima, especially for nonlinear or integer cases. Thus, approximate approaches such as piecewise linearization (Elimam et al., 1989), evolutionary algorithms (Wang et al., 2017; Bakhshipour et al., 2021), and topological-driven prediction-correction (Sitzenfrei et al.,

2020; Chegini and Li, 2022) are proposed to alleviate the computational burden. Nevertheless, a trade-off between solution quality and computational efficiency still exists for large-scale cases (Yu et al., 2024).





While SND and VNG share similar workflows, they serve fundamentally different purposes. SND optimizes network design for cost and performance, whereas VNG aims to reconstruct existing networks with incomplete information. This leads to a key challenge: while basic network attributes like terminal nodes, spatial layout, and pipe sizes can be possibly obtained through field surveys and engineering drawings, critical hydraulic parameters such as flow directions and invert elevations often remain unknown. These parameters are essential for accurate flood simulation but challenging to estimate at city scale. To address this gap, we propose a graph reconstruction approach that combines partial topological information with surface elevation data to better represent sewer drainage effects in urban flood modeling, thereby reducing simulation uncertainty.

In the remainder of this paper, we first describe the proposed workflow for sewer network reconstruction (Sect. 2) along with the corresponding 1D/2D coupled hydrologic-hydrodynamic model employed for urban pluvial flood simulation (Sect. 3) and then conduct a comparative analysis of the reconstructed models' performance under varying levels of information completeness, taking into account factors such as regulated facility removal and road-based layout simplification (Sect. 4).

## 2 Graph-based sewer network reconstruction

### 2.1 Graph representation of sewer networks

We represent sewer networks as graphs to leverage graph theory for topological analysis, providing a foundation for the reconstruction process. Given the spatial distribution of pipes, the sewer network can be abstracted as an undirected graph $G = (V, E)$ where $V = \{v_i\}$ and $E = \{e_{i,j}\}$ represent the sets of nodes and links, respectively (Shi Xiaoyu and Haifeng (2023)). When link directions are specified, e.g., by gravitational flow directions, the original undirected graph can be converted into the corresponding directed form $G_D = (V, E_D)$ where $E_D = \{\overrightarrow{e}_{i,j}\}$ represents the set of directed links evaluated as follows:

$$\overrightarrow{e}_{i,j} = \begin{cases} 1, & v_j \in \mathcal{N}_{G_D}(v_i) \\ 0, & v_j \notin \mathcal{N}_{G_D}(v_i) \end{cases}, \forall v_j \in \mathcal{N}_G(v_i) \tag{1}$$

where $\mathcal{N}_{G_D}(v_i)$ and $\mathcal{N}_G(v_i)$ represent the neighboring nodes of node $v_j$ in $G_D$ and $G$.

According to the number of directed links pointed from/to $v_i$, its out-degree $\mathrm{d}_{\mathrm{out},i}$ and in-degree $\mathrm{d}_{\mathrm{in},i}$ can be defined as follows:

$$\mathrm{d}_{\mathrm{out},i} = \sum_{v_j \in \mathcal{N}_G(v_i)} \overrightarrow{e}_{i,j}, \mathrm{d}_{\mathrm{in},i} = \sum_{v_j \in \mathcal{N}_G(v_i)} \overrightarrow{e}_{j,i}, \tag{2}$$

Besides these basic topological attributes, some useful structures can be derived from the graph representation. Considering the potential cycles in the geometric layout of $G$, the cycle basis $\mathcal{C} = \{C_k\}$ can be identified such that every cycle in $G$ can be expressed as the sum of cycles in $\mathcal{C}$ (Paton (1969)). For a cycle $C_k \in \mathcal{C}$, we group its internal nodes and links into the





corresponding sets denoted by $\mathrm{CV}_k$ and $\mathrm{CE}_k$, respectively. Thus, its out-degree $\mathrm{dC}_{\mathrm{out},k}$ and in-degree $\mathrm{dC}_{\mathrm{in},k}$ can be further

defined as follows:

$$\mathrm{dC}_{\mathrm{out},k} = \sum_{v_i \in \mathrm{CV}_k} \sum_{v_j \in \mathcal{N}_G(v_i) \backslash \mathrm{CV}_k} \overrightarrow{e}_{i,j}, \mathrm{dC}_{\mathrm{in},k} = \sum_{v_i \in \mathrm{CV}_k} \sum_{v_j \in \mathcal{N}_G(v_i) \backslash \mathrm{CV}_k} \overrightarrow{e}_{j,i} \tag{3}$$

where $\mathcal{N}_G(v_i) \backslash \mathrm{CV}_k$ represents the difference set between $\mathcal{N}_G(v_i)$ and $\mathrm{CV}_k$, i.e., the set of nodes that are neighboring nodes of $v_i$ but not in $C_k$.

According to Eq. 3, we define a cycle $C_k$ as an "island" if the sum of its in-degree and out-degree is equal to 1 and further

denote the set of nodes located in the islands of $G$ as $\mathrm{ISD}(G)$:

$$\mathrm{ISD}(G) = \bigcup_{C_k} \mathrm{CV}_k \quad \text{s.t.} \quad \mathrm{dC}_{\mathrm{out},k} + \mathrm{dC}_{\mathrm{in},k} = 1 \tag{4}$$

In addition to general properties, we also consider some sewer-specific characteristics for the convenience of following analysis. For a node $v_i$, we include it in the set of outfalls (denoted as $\mathcal{O}$) if its out-degree is 0, or in the set of sources (denoted as $\mathcal{S}$) if its in-degree is 0:

$$\mathcal{O} = \{v_i \mid v_i \in V, \mathrm{d}_{\mathrm{out},i} = 0\}, \mathcal{S} = \{v_i \mid v_i \in V, \mathrm{d}_{\mathrm{in},i} = 0\} \tag{5}$$

We also attach the invert elevation attribute to $v_i$ (denoted as $z_i$) which can be initialized as follows:

$$z_i^{(0)} = z_{g,i} - D_{\min} \tag{6}$$

where $z_i^{(0)}$ is the initial value of $z_i$; $z_{g,i}$ is the surface elevation at the location of $v_i$; $D_{\min}$ is the minimum allowable cover depth and set as $D_{\min} = 0.6\,\mathrm{m}$ by default (MOHURD (2021)).

By comparing invert elevations of neighboring nodes, link directions can be initialized as follows:

$$\overrightarrow{e}_{i,j}^{(0)} = \begin{cases} 1, & z_i^{(0)} \geq z_j^{(0)} \\ 0, & z_i^{(0)} < z_j^{(0)} \end{cases} \tag{7}$$

Fig. 1 shows an illustrative example of a simple sewer system in its graph representation with aforementioned properties.

## 2.2 Gravitational flow direction derivation

Since flow direction initialization using Eq. 7 does not consider the necessary topological constraints for a feasible layout

of the sewer network (Fig. 2), it may produce an unfeasible or unreasonable layout and therefore needs further correction (Reyes-Silva et al. (2023)).



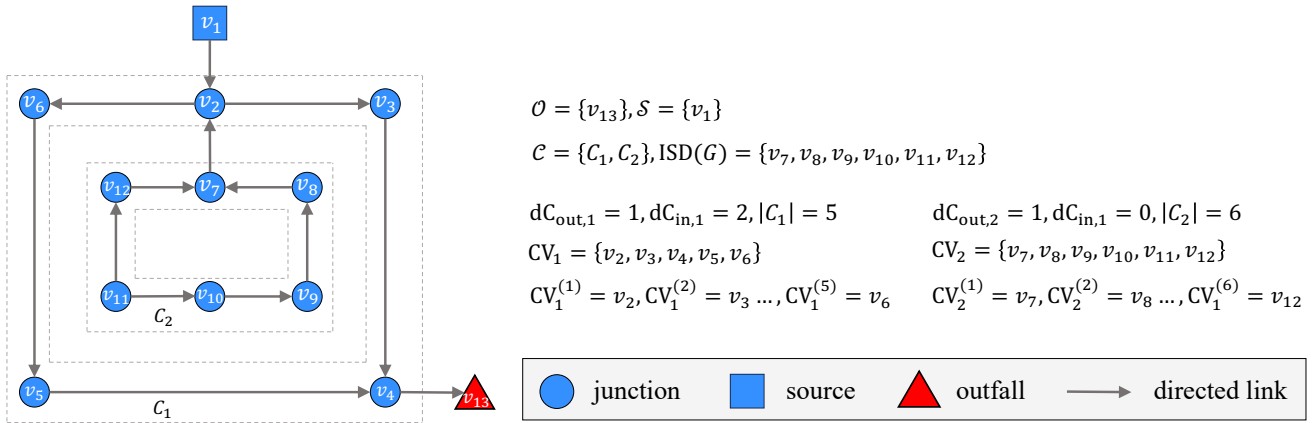

**Figure 1.** Graph representation of the sewer network.

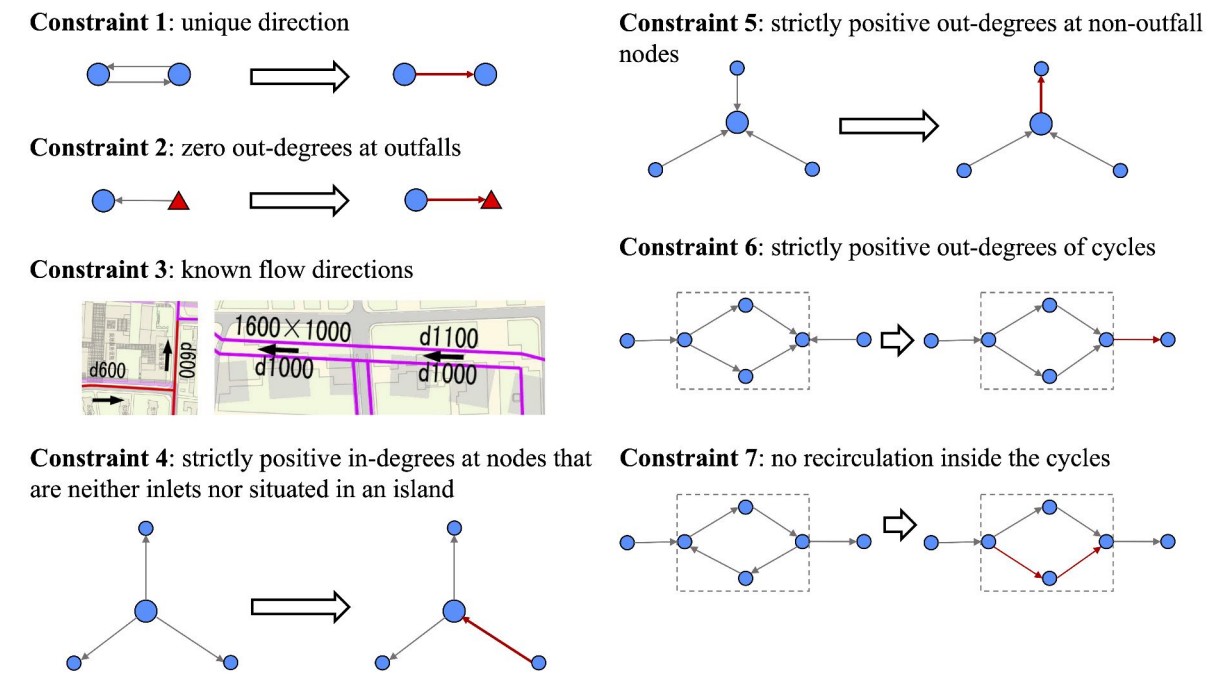

**Figure 2.** Topological constraints for a feasible sewer network.



In order to maintain flow directions as intact as possible during correction, we formulate the following 0-1 programming problems with linear constraints:

$$\min_{e_{ij}} \sum_{\overrightarrow{e}_{i,j}} \left| \overrightarrow{e}_{i,j} - \overrightarrow{e}_{i,j}^{(0)} \right| \tag{8a}$$


$$\text{s.t.} \begin{cases} \overrightarrow{e}_{i,j} = 1 - \overrightarrow{e}_{j,i} & \text{(8b)} \\[4pt] \overrightarrow{e}_{i,j} = 1, \forall v_j \in \mathcal{O} & \text{(8c)} \\[4pt] \overrightarrow{e}_{i,j} = \overrightarrow{e}_{i,j}^{*}, \forall \overrightarrow{e}_{i,j} \in \mathcal{E}_D^{*} & \text{(8d)} \\[4pt] d_{\text{in},i} \geq 1, \forall v_i \notin \mathcal{I} \cup \text{ISD}(G) & \text{(8e)} \\[4pt] d_{\text{out},i} \geq 1, \forall v_i \notin \mathcal{O} & \text{(8f)} \\[4pt] dC_{\text{out},k} \geq 1, \forall C_k \in \mathcal{C} & \text{(8g)} \\[4pt] \overrightarrow{e}_{\text{CV}_k^{(|C_k|)},\text{CV}_k^{(1)}} + \displaystyle\sum_{i=1}^{|C_k|-1} \overrightarrow{e}_{\text{CV}_k^{(i)},\text{CV}_k^{(i+1)}} \leq |C_k| - 1, \forall C_k \in \mathcal{C} & \text{(8h)} \end{cases}$$

where $|C_k|$ is the number of nodes in $C_k$; $\mathcal{E}_D^{*}$ is the set of links with known directions (denoted by $\overrightarrow{e}_{i,j}^{*}$); $\overrightarrow{e}_{\text{CV}_k^{(|C_k|)},\text{CV}_k^{(1)}} + \sum_{i=1}^{|C_k|-1} \overrightarrow{e}_{\text{CV}_k^{(i)},\text{CV}_k^{(i+1)}}$ is the sum of consecutively connected direction values along $C_k$.

The rationale for each constraints can be summarized as follows (cf. Fig. 2):

- Eq. 8b: No bi-directional links between consecutive nodes

- Eq. 8c: Outfalls must have zero out-degree

- Eq. 8d: Preserves known link directions

- Eq. 8e: Non-inlet and non-island nodes must have positive in-degrees (Eq. 4)

- Eq. 8f: Non-outfall nodes must have positive out-degrees

- Eq. 8g: Cycles must have positive out-degrees for flow passage

- Eq. 8h: No recirculation within cycles

Prior to optimization, pipes are discretized into 50 m segments following common inlet spacing (MOHURD, 2021). This ensures uniform link lengths for objective evaluation and provides computational elements with inlets at endpoints for surface-sewer exchange in hydraulic simulation.





### 2.3 Nodal invert elevation derivation

Corrected gravitational flow directions pave the way for the derivation of nodal invert elevation prepared for downstream hydraulic simulation. Based on invert elevation initialized by Eq. 6, further adjustment is needed to keep the slopes of links within the range of minimum and maximum allowable slopes (Safavi and Geranmehr, 2017).

In order to achieve the least deviation from initial results considering sewer construction costs proportional to excavation depth (Swamee and Sharma, 2013; Machine Hsie and Huang, 2019), we introduce the following nonlinear optimization problem to estimate the invert elevation of nodes:

$$
\min_{z_i} \sum_{z_i} \left| z_i - z_i^{(0)} \right|
$$
$$
\text{s.t.} \quad \begin{cases} i_{i,j} \leq -i_{\min,i,j}, & \forall \overrightarrow{e}_{i,j} = 1 \\ i_{i,j} \geq -i_{\max,i,j}, & \forall \overrightarrow{e}_{i,j} = 1 \end{cases} \tag{9}
$$

where $i_{i,j}$ is the slope of $\overrightarrow{e}_{i,j}$; $i_{\min,i,j}, i_{\max,i,j}$ are the minimum and maximum allowable slopes required for $\overrightarrow{e}_{i,j}$ according to Table 1 summarized from the Standard for Design of Outdoor Wastewater Engineering (MOHURD, 2021), considering both explicitly stated slope limits and those derived from velocity constraints (Swamee and Sharma, 2013).

For simplicity but without loss of generality, $i_{i,j}$ is estimated using the invert elevation of two consecutive nodes, ensuring the linearity of constraints for the above optimization problem:

$$
i_{i,j} = -\frac{1}{d_{i,j}} z_i + \frac{1}{d_{i,j}} z_j \tag{10}
$$

It should be also noted that Eq. 10 can be also extended to the case of multiple consecutive nodes and corresponding slope estimator still remains in the form of the linear combination of nodal invert elevation (see Appendix A for details), which maintains the linearity of constraints in the original optimization problem.

### 2.4 Computational efficiency considerations

The objective functions of Eq. 8 and Eq. 9 are expressed in the form of the sum of absolute deviations and can be further reduced into an equivalent linear form as follows (Wagner (1959); Giangrande et al. (2013), thus ensuring their solution efficiency for large-scale applications.

Without loss of generality, for minimizing the summation of absolute values:

$$
L = \sum_{i=1}^{N} |k_i x_i - b_i| \tag{11}
$$

where $k_i, b_i$ are fixed numbers with respect to $N$ decision variables $x_i$, we can introduce auxiliary decision variables $z_i = |k_i x_i - b_i|$ constrained by:





**Table 1.** Slope constraints with respect to different sizes according to Standard for Design of Outdoor Wastewater Engineering (MOHURD, 2021).

| Diameter [mm] | Minimum slope | Maximum slope |
|---|---|---|
| 300 | 0.002 | 0.41 |
| 400 | 0.0015 | 0.28 |
| 500 | 0.0012 | 0.21 |
| 600 | 0.001 | 0.16 |
| 800 | 0.0008 | 0.11 |
| 1000 | 0.0006 | 0.08 |
| 1200 | 0.0006 | 0.06 |
| 1400 | 0.0005 | 0.05 |
| > 1500 | 0.0005 | 0.05 |

$$
\begin{cases}
z_i \geq k_i x_i - b_i \\
z_i \geq -k_i x_i + b_i \\
z_i \geq 0
\end{cases}
\tag{12}
$$

Thus, the original unconstrained nonlinear problem can be reformulated into an equivalent linear one with additional constraints:

$$
\min_{x_i, z_i} \sum_{i=1}^{N} z_i
$$
$$
\text{s.t.} \quad
\begin{cases}
z_1 \geq k_i x_1 - b_0, z_1 \geq -k_i x_1 + b_0, z_1 \geq 0 \\
..., ... \\
z_N \geq k_i x_N - b_N, z_N \geq -k_i x_N + b_N, z_N \geq 0
\end{cases}
\tag{13}
$$

Furthermore, if additional constraints are present with Eq. 11, they can be added directly to the reformulated problem (Eq. 13) without loss of equivalence. This reformulation strategy has been used in previous research (Giangrande et al., 2013) and thus can be easily extended to our work for efficient sewer network reconstruction (Eq. 8 and Eq. 9).

## 3 AUTOSHED: A 1D/2D coupled hydrologic-hydrodynamic model

Given the sewer network reconstructed from incomplete information, we further adopt a 1D/2D coupled hydrologic-hydrodynamic model named AUTOSHED to simulate the pluvial flood process in the urbanized area. AUTOSHED delineates the whole simulation domain into multiple triangular-shaped units (TSUs). For each unit, AUTOSHED uses the Rain-on-Grid approach, also





known as the Fully Hydrodynamic Approach (Hall, 2015; Chen and Huang, 2024; Perrini et al., 2024), and performs sequential hydrologic-hydrodyamic computations for surface runoff generation and routing, respectively.

## 3.1 Surface runoff generation and routing

The surface runoff generation scheme is mainly built upon the core module of the Surface Urban Energy and Water Balance
Scheme (SUEWS) (Järvi et al., 2011) and Tsinghua Integrated Hydrological Modeling System (THIHMS) (Ni et al., 2008) except for overland and channel flow routing. Each TSU is divided into impervious and pervious parts according to the empirical impervious ratio $\alpha_{\mathrm{imp}}$ derived from the corresponding land use types in the unit using area-weighted averaging, which is also known as the mosaic method (Cao et al., 2020; Krebs et al., 2014). For the impervious part, surface runoff $\Delta S_{\mathrm{imp}}$ is equal to rainfall. For the pervious part, surface runoff $\Delta S_{\mathrm{p}}$ equals rainfall minus vegetation interception, building interception and soil
infiltration, parameterized using leaf area index, building footprint ratio and soil properties (Järvi et al., 2011; Ni et al., 2008). The total surface runoff $\Delta S_{\mathrm{sr}}$ is calculated as the area-weighted sum of $\Delta S_{\mathrm{p}}$ and $\Delta S_{\mathrm{imp}}$:

$$\Delta S_{\mathrm{sr}} = \alpha_{\mathrm{imp}}\Delta S_{\mathrm{imp}} + (1 - \alpha_{\mathrm{imp}})\Delta S_{\mathrm{p}} \tag{14}$$

After obtaining total surface runoff, a well-balanced formulation of 2D shallow water equations (SWEs) (Song et al., 2011) is adopted to simulate the subsequent surface routing process:

$$\frac{\partial \mathbf{U}}{\partial t} + \frac{\partial \mathbf{E}}{\partial x} + \frac{\partial \mathbf{G}}{\partial y} = \mathbf{S} \tag{15}$$

where

$$\mathbf{U} = \begin{bmatrix} h \\ hu \\ hv \end{bmatrix}, \quad \mathbf{E} = \begin{bmatrix} hu \\ hu^2 + \mathrm{g}(h^2 - z_b^2)/2 \\ huv \end{bmatrix}, \quad \mathbf{G} = \begin{bmatrix} hu \\ huv \\ hv^2 + \mathrm{g}(h^2 - z_b^2)/2 \end{bmatrix},$$

$$\mathbf{S} = \begin{bmatrix} \Delta S_{\mathrm{sr}} - q_{\mathrm{sd}} \\ -\mathrm{g}(h + z_b)\frac{\partial z_b}{\partial x} - ghS_{fx} \\ -\mathrm{g}(h + z_b)\frac{\partial z_b}{\partial y} - ghS_{fy} \end{bmatrix} \tag{16}$$

where $h$ is the water depth; $u, v$ are the depth-averaged velocity components in the x- and y-directions; $z_b$ is the bed elevation; g is the gravitational acceleration; $q_{\mathrm{sd}}$ is the mass loss term caused by sewer drainage; $S_{fx}, S_{fy}$ are the friction slopes in the x-
and y-directions which can be parameterized by the Manning formulae:

$$S_{fx} = \frac{n^2 u \sqrt{u^2 + v^2}}{h^{4/3}}, S_{fy} = \frac{n^2 v \sqrt{u^2 + v^2}}{h^{4/3}} \tag{17}$$





where $n$ is the empirical Manning coefficient initially derived from land use types using the mosaic method and further increased at TSUs intersected with buildings using the building roughness method (Schubert and Sanders, 2012):

$$n = n_0 + (0.5 - n_0) \cdot b_f \tag{18}$$

where $n, n_0$ are the final and initial estimates of the empirical Manning coefficient and $b_f$ is the intersected building area ratio at the TSU.

Considering the significance of topography in simulating pluvial flood, AUTOSHED utilizes a sloping bottom model to define the bed elevation at three vertices of each TSU (Begnudelli and Sanders, 2006), thereby achieving the second-order spatial accuracy in surface topography representation. AUTOSHED uses the finite volume method to solve the above 2D

SWEs where the MUSCL reconstruction is used for estimating flow variables at the midpoints of unit edges, the HLLC approximate Riemann solver is used for evaluating mass and momentum fluxes across edges, the two-stage explicit Runge-Kutta approach is used for updating flow variables. Details of methodologies on solving 2D SWEs can be found in the work of Song et al. (2011). The modules of surface runoff generation and routing are implemented in C++ and CUDA (Computed Unified Device Architecture), which can leverage many-core parallel capacity of the graphics processing units to significantly

boost the computation performance of refined coupled hydrologic-hydrodynamic simulation with millions of units (Lacasta et al. (2014)).

## 3.2 Sewer flow simulation and its coupling with surface flow

In the TSU with rainwater inlets, vertical linkages are established for bidirectional coupling between the surface flow and the sewer flow. A modified form of the 1D Saint-Venant equations is introduced to represent the sewer flow process using a

node-link approach (Rossman and Huber (2017)):

$$\begin{cases} \dfrac{\partial H}{\partial t} = \dfrac{Q_{1D-2D} + \sum Q_{nl}}{A_{as}} \\ \dfrac{\partial Q}{\partial t} = 2U\dfrac{\partial A}{\partial t} + U^2\dfrac{\partial A}{\partial x} - gA\dfrac{\partial H}{\partial x} - gAS_f \end{cases} \tag{19}$$

where $H$ is the total nodal hydraulic head of water; $Q$ is the flow rate; $U$ is the flow velocity; $A$ is the cross-sectional area; $A_{as}$ is the node assembly surface area consisting of node's storage surface area and half the surface area of connected links (Rossman (2015)); $S_f$ is the friction slope parameterized by the Manning formula; $Q_{nl}$ are the flow contributed by neighboring

links in the sewer system; $Q_{1D-2D}$ is the flow sourced from the mass exchange between the surface flow and the sewer flow.

Considering the complexity in flow exchange between the surface and underground sewer system (Dai et al. (2023)), AUTOSHED follows previous research works and adopts the weir-/orifice-based empirical formulae to parameterize $Q_{1D-2D}$ at each inlet (Buttinger-Kreuzhuber et al. (2022)). Assuming $Q_{1D-2D}$ is positive when water flows from the TSU to the linked inlet, the corresponding parameterization formulae can be summarized as follows:





$$
\text{210} \quad Q_{\text{1D}-\text{2D}} = \begin{cases} \dfrac{2}{3} c_w P \sqrt{2g} h_{2\text{D}}^{1.5}, & H_{1\text{D}} < z_{\text{bc}} \\[2mm] c_{o,s} A \sqrt{2g} [h_{2\text{D}} - (H_{1\text{D}} - z_{\text{bc}})]^{0.5} & z_{\text{bc}} \leq H_{1\text{D}} < z_{\text{bc}} + h_{2\text{D}} - A/P \\[2mm] c_{w,s} P \sqrt{2g} h_{2\text{D}} [h_{2\text{D}} - (H_{1\text{D}} - z_{\text{bc}})]^{0.5} & z_{\text{bc}} + h_{2\text{D}} - A/P \leq H_{1\text{D}} < z_{\text{bc}} + h_{2\text{D}} \\[2mm] - c_{w,s} P \sqrt{2g} h_{2\text{D}} [(H_{1\text{D}} - z_{\text{bc}}) - h_{2\text{D}}]^{0.5} & z_{\text{bc}} + h_{2\text{D}} \leq H_{1\text{D}} < z_{\text{bc}} + h_{2\text{D}} + A/P \\[2mm] - c_{o,s} A \sqrt{2g} [(H_{1\text{D}} - z_{\text{bc}}) - h_{2\text{D}}]^{0.5} & H_{1\text{D}} \geq z_{\text{bc}} + h_{2\text{D}} + A/P \end{cases} \tag{20}
$$

where $H_{1\text{D}}$, $h_{2\text{D}}$ and $z_{\text{bc}}$ are the total hydraulic head at the inlet, the water depth at the TSU and the center bed elevation at the TSU, respectively; $A$ and $P$ are the inlet's area and perimeter, respectively; $c_w$, $c_{w,s}$ and $c_{o,s}$ are discharge coefficients for the free weir, the submerged weir, and the orifice equations, respectively. By default, AUTOSHED sets $c_w = 0.56$, $c_{w,s} = 0.11$ and $c_{o,s} = 0.2$ (Rubinato et al. (2017)) and $A = 0.3375 \text{ m}^2$, $P = 2.4 \text{ m}^2$ assuming that each inlet is rectangular whose size is

$0.45 \text{ m} \times 0.75 \text{ m}$ according to field investigation. Considering that one TSU can be linked with multiple inlets, we have:

$$
q_{\text{sd}} = \frac{1}{A_{\text{T}}} \sum_{i=1}^{N} Q_{\text{1D}-\text{2D}}^{(i)} \tag{21}
$$

where $A_{\text{T}}$ is the total area of the TSU, $N$ is the total number of linked inlets and $Q_{\text{1D}-\text{2D}}^{(i)}$ is the sewer drainage rate at the $i$-th inlet derived from Eq. 20 and further constrained by the water availability of both the TSU and the inlet (Buttinger-Kreuzhuber et al. (2022)):

$$
\text{220} \quad Q_{\text{1D}-\text{2D}}^{(i)} = \min \left( \frac{\min(h_{2\text{D}} A_T, V_{\text{as}})}{\Delta t} \frac{Q_{\text{1D}-\text{2D}}^{(i)}}{\sum_{i=1}^{N} Q_{\text{1D}-\text{2D}}^{(i)}}, Q_{\text{1D}-\text{2D}}^{(i)} \right) \tag{22}
$$

where $V_{\text{as}}$ is the water volume at the inlet assembly determined by $A_{\text{as}}$ and $H_{1\text{D}}$; $\Delta t$ is the coupling time step evaluated as the minimum value between the timesteps of the 1D and 2D solvers (Chen et al. (2018)).

    In order to solve Eq. 19 jointly with Eq. 16, AUTOSHED utilizes the 1D routing portion of the Storm Water Management Model (SWMM) 5.1.015 (Rossman (2015)) and implements communication routines for 1D/2D state variables such as water

depth and hydraulic head (Eq. 20) via interface functions by compiling SWMM as a dynamic link library (Leandro and Martins (2016)).

    In addition to unregulated nodes and links, we also account for the controlling effects of pump stations and wastewater treatment plants using Type 4 pumps (whose flow rate changes linearly with the inlet node depth until the maximum pumping capacity is reached) and nodes with limited maximum inflows in SWMM, respectively (Rossman (2015)).

## 3.3 Study area and data

The study area is the main city zone of Yinchuan (CYC), Ningxia Hui autonomous region, situated in the northwestern part of China, covering an area of $137.15 \text{ km}^2$. Desipte being located in an arid region with an average annual rainfall of 189 mm and



annual evaporation of 1825 mm, CYC has experienced several flood disasters due to increasing extreme rainfall events in recent years (Lu et al. (2024)). According to local documents, most of the drainage system in CYC is the combined sewer system, and therefore its drainage capacity is largely constrained by downstream wastewater treatment plants (WTPs) due to restrictive environmental regulations, which may result in overflow at rainwater inlets and further amplify the surface inundation risks.

In order to establish the urban pluvial flood model for scenario analysis, we collect multi-source geographical datasets listed as follows:

- 5 m digital elevation model (Fig. 3 (a)) generated from the stereo images of ZiYuan-3 satellite.

- 10 m land use map (Fig. 3 (b)) collected from the European Space Agency (ESA) WorldCover 2021 product (Zanaga et al. (2022)) and covers tree, grassland, cropland, built-up and bareland area in the study area.

- Building footprint (Fig. 3 (a)) and road network layers (Fig. 3 (b)) downloaded via the application programming interface of the Baidu map. Considering potential underestimation of impervious area identified from remote sensing datasets (Weng (2012)), we further modify the land use type as built-up area at pixels intersected with buildings and roads.

- Sewer network layout and corresponding attribute of pipeline sizes (Fig. 4 (a)) digitized from investigation and design documents authorized by the local government. It should be noted that when detailed information is unavailable, previous research would generate a virtual sewer network from the street network (Montalvo et al. (2024)). In order to investigate potential effects induced by such simplification, we further create a road-based sewer network by identifying pipes intersected with the major roadway zone and then performing necessary reconnections at pipes broken intermediately (Fig. 4 (b)). Here we define the term of "major road" (Fig. 3 (b)) as the union of the primary, secondary, tertiary and trunk roads and then generate corresponding roadway zones by buffering road centerlines with the width evaluated from The Design of Urban Road Engineering (CJJ 37—2012).

The formula of the designed rainfall in CYC is also collected based on local rainfall statistics:

$$q = \frac{551.4(1 + 0.584 \lg P)}{(t + 11)^{0.669}} \tag{23}$$

where $q$ is the rainfall intensity, L/(s · ha); $P$ is the return period; $t$ is the rainfall duration, min.

Seven designed rainfalls with return years of 1a, 5a, 10a, 20a, 50a, 100a, and 200a are selected for the scenario analysis where each rainfall event lasts for 2 h (Fig. 5), followed by an additional 2 h period without any precipitation.

### 3.4 Baseline configurations

Four baselines are selected for the comparison of the performance between the proposed approach (denoted as the FSR approach) and other approximation methods:





**Figure 3.** Geographical datasets in the study area. (a) Digital elevation model. (b) Land use map and road networks. Basemaps are provided by Esri.





**Figure 4.** Sewer network in the study area. (a) Complete sewer network digitized from the local document. (b) Simplified sewer network filtered within major roadway zone. Basemaps are provided by Esri.

- **F**ull-**S**ewer-reconstruction-**N**o-controls (FSN) which uses the same reconstruction approach but ignores the existence of regulated facilities such as pump stations and wastewater treatment plants. To be specific, regulated facilities located at outflow and intermediate points are configured as free outfalls and uncontrolled junctions, respectively.





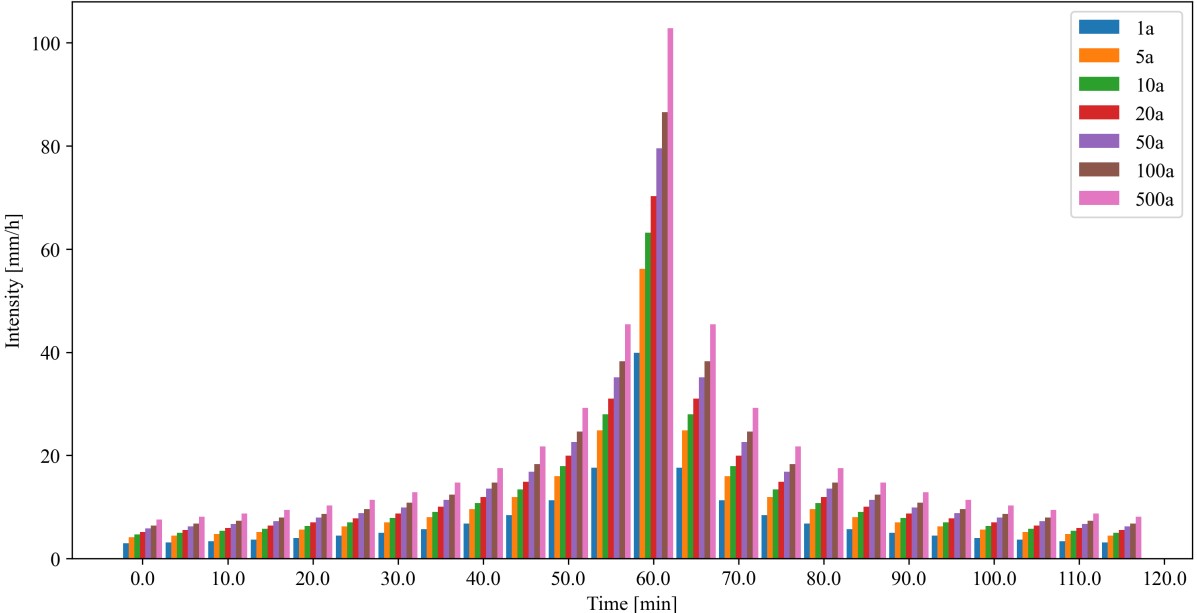

**Figure 5.** 2-hour design rainfall scenarios with varying return periods (1a, 5a, 10a, 20a, 50a, 100a and 500a).

- **R**oad-based-**S**ewer-reconstruction-**N**o-controls (RSN) which uses the same reconstruction approach but adopts the layout of sewers filtered by the major roadway zone (Fig. 4 (b)) and ignores the existence of regulated facilities.

- **R**oad-**D**rainage **A**pproximation (RDA, Xing et al. (2021); Li et al. (2023)) which subtracts the mass at a uniform rate $i_d$ from the roadway zone buffered from road centerlines using widths evaluated by The Design of Urban Road Engineering (CJJ 37—2012).

- **N**o-**D**rainage **A**pproximation (NDA) which ignores the drainage effects of the sewer network.

According to whether a sewer hydraulic module is reconstructed for simulation with AUTOSHED, we can further classify the above approximation methods into two categories: physical approaches (including FSR, FSN and RSN) and equifinal ones (including RDA and NDA).

## 4 Performance Analysis: From Graph Reconstruction to Urban Flood Simulation

### 4.1 Sewer network reconstruction performance

According to the layout of the digitized sewer network (Fig. 4 (a)), gravitational flow directions are initialized by surface elevation (Eq. 7) and then estimated by solving 0-1 programming problems (Eq. 8). For simplicity, Fig. 6 summarizes the corresponding results for the initialized and reconstructed directions of the complete sewer network with 11 outfall points



consisting of wastewater treatment plants and pumping stations. The directions of 359 in 908 pipes are reversed to satisfy the topological constraints. The majority of modified pipes are located in the southern part of the study area and exhibit a pattern

of cyclic structures over a relatively flat areas, posing challenges on the direct identification of dominant flow directions and thus necessitating the incorporation of additional topological constraints (Eq. 8b-Eq. 8h) for rational reconstruction.

According to reconstructed flow directions, nodal invert elevation is further derived from values initialized by a constant cover depth (Eq. 6) and the slope distributions before / after adjustment are summarized in Fig. 6. The initial slopes of both complete and simplified sewer network exhibit a relatively symmetric distribution centered on 0, indicating adverse slopes

in nearly half a number of links due to nearly reversed 40% pipes in the previous step. After reconstruction using Eq. 9, slope distributions are shifted into strictly positive ones where the majority of slopes are around 0.001. Additionally, the simplified sewer network shares a similar pattern in the slope distribution with the complete one except for minor increases around 0 because its layout is slightly modified due to aforementioned reconnection of broken pipes in Sect. 3.3. Thus, it can be concluded that the proposed reconstruction approach is robust to empirical simplification of the sewer network layout

following road networks and can be stably extended to cases using street-based network generation.

Owing to linearized programming formulation used in the process of gravitational flow directions and nodal invert elevation reconstruction (Sect. 2.4), 1D sewer hydraulic models can be built from prepared geographical datasets in a few minutes, resulting in 8242 links and 8131 inlets (initialized with a uniform spacing of 50 m; cf. Sect. 2.2) in FSR/FSN, 5268 links and 5188 inlets in RSN. Furthermore, the entire study area is discretized into 7222150 TSUs with a median area of 18.2 $\mathrm{m}^2$ for

surface runoff generation and routing computation.

### 4.2 Model calibration and evaluation

To facilitate the comparison of pluvial flood simulation performance using different approaches, we first calibrate the model parameters with a severe storm hitting CYC on 11 July 2022 (denoted as the 20220711 storm). This storm brings an average total rainfall of 57.8 mm and a peak intensity of 44.5 mm/h, resulting in 31 inundated points reported by the local government,

of which 18 are recorded with corresponding maximum waterlogging depth. Considering its representativeness as disastrous torrential events and data availability, we use the information of recorded maximum waterlogging depth to calibrate the parameters in the aforementioned model and related approximation approaches for the sewer drainage effect.

For AUTOSHED, the main physical parameters include the empirical impervious ratio $\alpha_{\mathrm{imp}}$ and the Manning coefficient $n$ estimated from the types of land use. Considering the dominance of the built-up area and attempts to avoid possible parameter

equifinality between physical process such as soil infiltration and sewer drainage, we perform simple tests of model performance using recommended empirical values of corresponding parameters from the Techical Specification for Construction and Application of Mathematical Model of Urban Flooding Prevention and Control System (DB11/T 2074-2022) and finalize the values of parameters in Table. 2 without exhaustive search. Fig. 8 shows the inundation map simulated by AUTOSHED with the sewer hydraulic module reconstructed from the FSR approach under the 20220711 storm.

The coefficient of determination ($\mathrm{R}^2$) is used to quantify the overall performance of the models using different parameters:



**Figure 6.** Comparison of initial and reconstructed gravitational flow directions of the complete sewer network in the study area. (a) Flow directions initialized by surface elevation. (b) Flow directions reconstructed from topological constraints. Basemaps are provided by Esri.

$$R^2 = 1 - \frac{\sum_{i=1}^{N} \left( \hat{h}_i - h_i \right)^2}{\sum_{i=1}^{N} \left( h_i - \mu_h \right)^2} \tag{24}$$





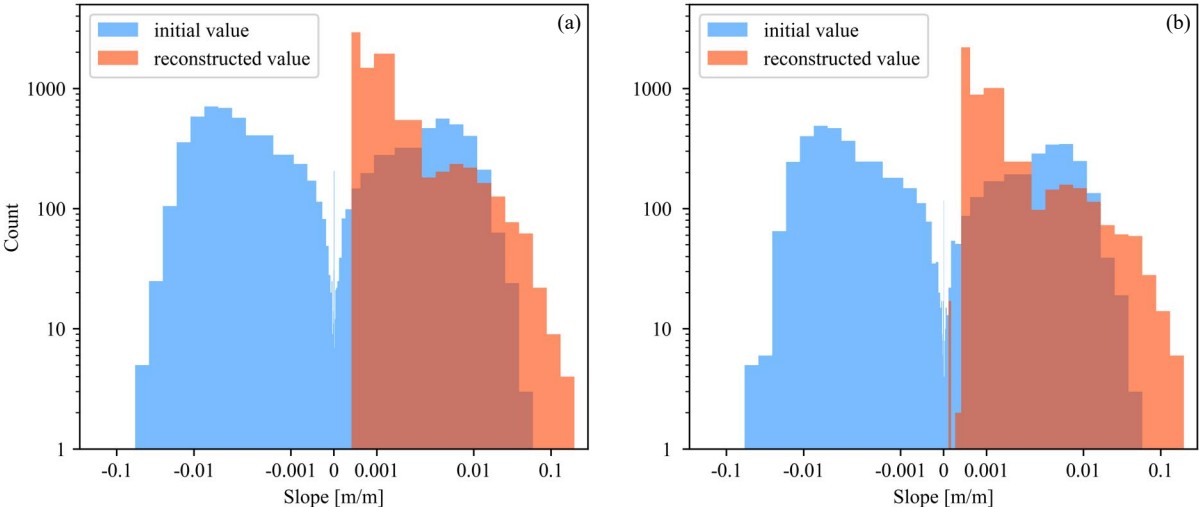

**Figure 7.** Comparison of initial and reconstructed sewer slope distributions. (a) Complete sewer network used in the FSR and FSN approaches. (b) Road-based sewer network used in the RSN approach.

**Table 2.** Calibrated parameters of the AUTOSHED model.

| Land use type | Impervious ratio [$\mathrm{m^2/m^2}$] | Manning coefficient [$\mathrm{s/m^{1/3}}$] |
|---|---|---|
| tree | 0.05 | 0.06 |
| grassland | 0.05 | 0.08 |
| cropland | 0.05 | 0.08 |
| built-up | 1.0 | 0.015 |
| bareland | 0.4 | 0.04 |
| water | 1.0 | 0.02 |

where $N = 18$ is the number of inundated points with known maximum depth, $\hat{h}_i$ and $h_i$ are the maximum depth simulated and observed, $\mu_h$ is the mean of the observed maximum depth.

According to Fig. 9 (a), the FSR approach achieves an $R^2$ of 0.95, which indicates good agreement between the simulated
and observed values and thus consolidates the physical background for further comparison.

Based on calibrated parameters of the physical model, we further use the grid search strategy to optimize the most possible equivalent drainage rate $i_d$ used in the RDA method when detailed sewer network information is unavailable. Specifically, we set the possible range of $i_d$ according to the local design drainage standard and perform simulations with $i_d$ = 2.5, 5.0, 7.5, 10.0, 12.5, 15.0 mm/h. According to Fig. 9 (b), $i_d$ = 2.5 mm/h achieves the highest $R^2$ and thus selected as the final value of
drainage rate for further scenario analysis.



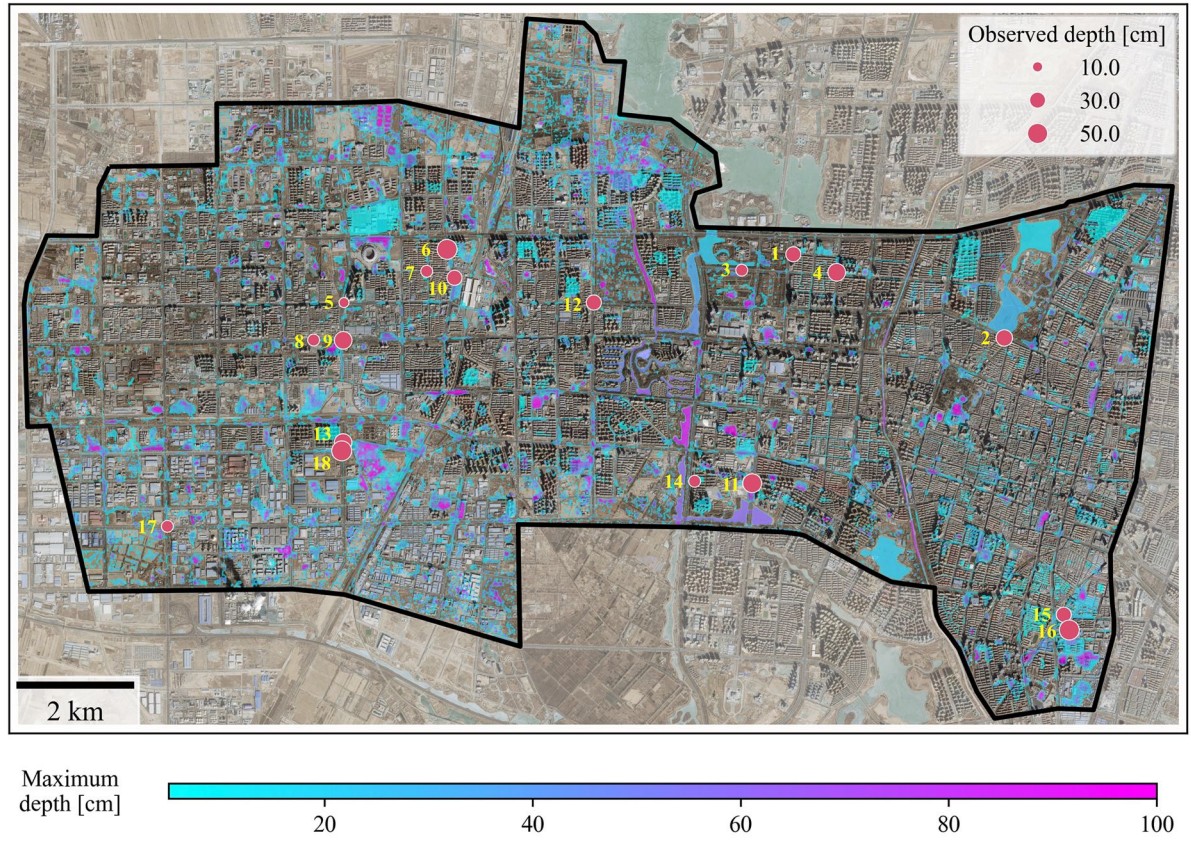

**Figure 8.** Maximum inundation depth map simulated by the FSR approach under the 20220711 storm. The basemap is provided by Esri.

## 4.3 Extreme rainfall analysis: both local inlets and global regulation matter

After calibrating the AUTOSHED model integrated with reconstructed sewer hydraulic modules, we now delve deeper into understanding how the proposed graph reconstruction-based approach performs compared to simplified alternatives across different rainfall scenarios and varying levels of available information. The aforementioned 20220711 storm event, with an

average total rainfall exceeding the 500-year return period threshold (40.8 mm), highlights its extremity in an arid region. According to Fig. 9, when information on regulated facilities and detailed layouts is unavailable, simplified physical approaches, such as FSN and RSN, exhibit relatively lower accuracy than calibrated equifinal ones, which approximate limited drainage capacities during extreme rainfall using a small equivalent drainage rate.

Considering physical approaches incorporate drainage effects at rainwater inlets, we further investigate the relationship

between simulation error in maximum inundation depth and the distances to the nearest inlets, denoted as $D_f$ for the complete sewer network and $D_s$ for the simplified one (see Sect. 3.3), respectively. As illustrated in Fig. 10, simulation error variability decreases as $D_f$ and $D_s$ increase, with the maximum absolute value of deviations remaining less than 5 cm for distances





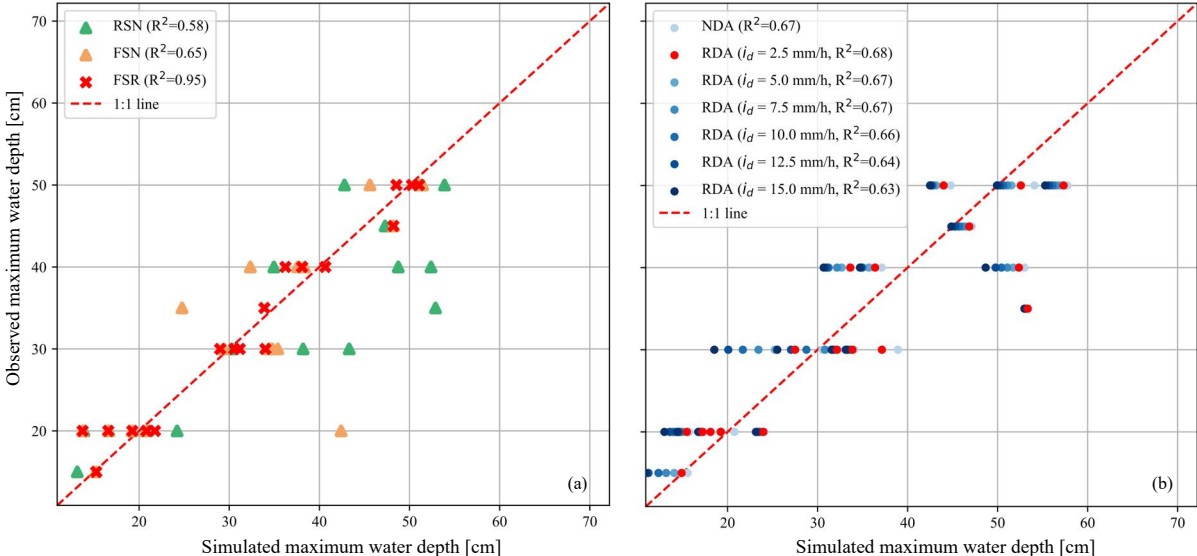

**Figure 9.** Simulation performance of the maximum inundation depth at inundated points using different approximation methods.

exceeding 30 m, where local microtopography predominantly influences the inundation. Furthermore, NDA can underestimate maximum inundation depths at points such as P8 and P18, suggesting the occurrence of potential overflows surrounding nearby
rainwater inlets due to insufficient drainage capacity under extreme rainfall.

To assess differences in simulated sewer drainage performance, we further summarize accumulated discharges at rainwater inlets and conduct two steps of comparisons according to available information.

  – **Effects of neglecting regulation:** when neglecting the information of regulated facilities, FSN significantly underestimates maximum inundation depths at points such as P2 and P9 compared to FSR (Fig. 10). This mainly results from
overestimated inflows ($\Delta \sum Q_{1D-2D} > 0$) at surrounding inlets, as seen at P2 in Fig. 11 (a) and (e). However, FSN also overestimates maximum inundation depths at certain points such as P1 and P3 (Fig. 10), which may seem counterintuitive. Analysis of P1's nearby inlets reveals lower inflows using FSN, compared to FSR, particularly at IL1 (Fig. 11 (b) and (f)), causing model overestimation of surrounding maximum inundation depth. After tracking accumulated discharges at downstream inlets with respect to IL1, we further locate the drainage bottleneck at one pump station PM1.
As illustrated in Fig. 11 (d), its removal can significantly increase inflows at upstream inlets, further raising water heads at adjacent sewer nodes and consequently reducing upstream inlet drainage capacities due to backwater effects.

  – **Impact of network simplification:** when approximating the sewer layout through major roads, RSN generally overestimates the maximum water depths at inundated points within 30 m of rainwater inlets compared to FSN (Fig. 10), driven by intensified overflows from excessive surface water flowing through a single drainage pathway, as seen at P1 in Fig.
12 (a) and (c). However, some inundated points like P13 and P18, where surrounding inlets are removed, may also ex-



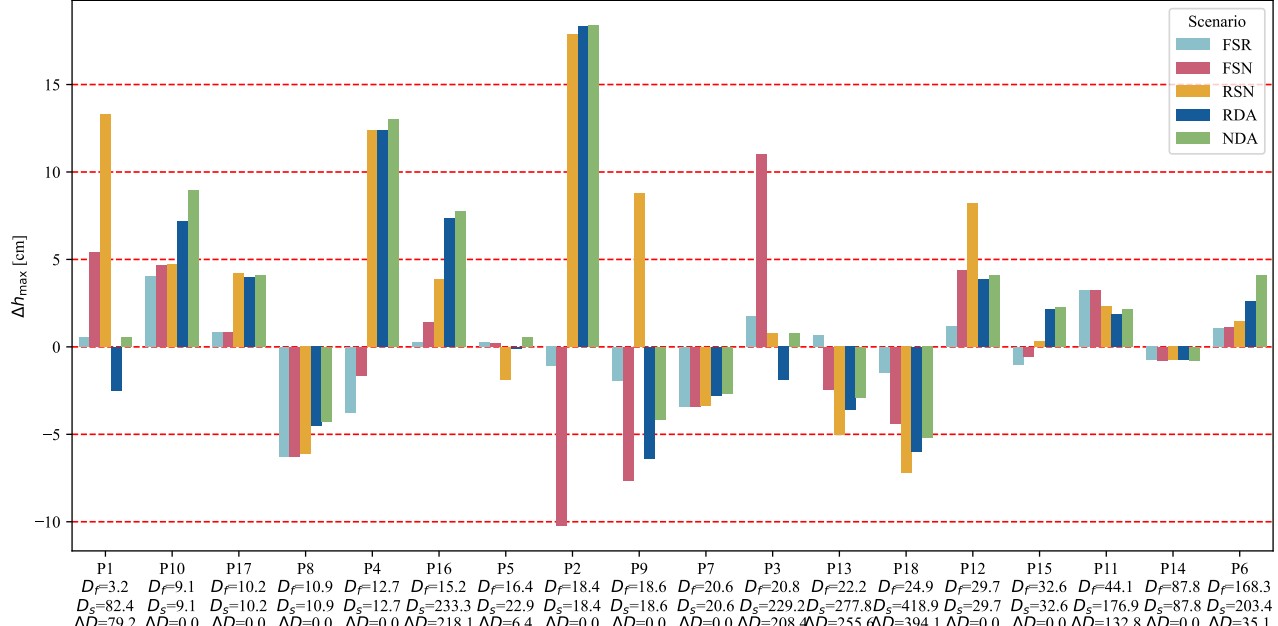

**Figure 10.** Simulation error of maximum inundation depth at inundated points where $\Delta h_{\max}$ stands for the simulated depth minus observed ones and $D_f$ and $D_s$ are the distances to the closest rainwater inlet of the complete sewer network used in the FSR, FSN approach and the simplified sewer network used in the RSN approach. $\Delta D = D_s - D_f$. The index of inundated points follow Fig. 8.

hibit underestimated maximum water depths. This occurs because these inlets initially contribute to surface overflows in FSN-based simulations, as evidenced by the underestimation of maximum water depth in Fig. 10 when NDA is adopted. In particular, we can also observe the overestimation of maximum water depths at certain flood points without removing local inlets during simplification, such as P2 and P4 with $\Delta D = 0$ in 10. Examining the simulated total discharges at rainwater inlets near P2 (Fig. 12 (b) and (d)) reveals that increased local inflows are outweighed by amplified upstream overflows, thus exacerbating downstream inundation conditions.

In summary, while physical approaches simplified through major roads can yield a similar pipe slope distribution (Sect. 4.1), achieving accurate simulation of maximum inundation depths still requires the proper configuration of regulated facilities and drainage pathways. Otherwise, equifinal methods, flexible in parameter calibration, can offer better performance, especially when observational data are available.

## 4.4 Design rainfall analysis: physical approaches outperform but require more complete information

In order to assess whether equifinal approaches always exhibit comparable or even better performance with physical counterparts, we further conduct design rainfall scenario analysis. The Probability of Detection (POD), False Alarm Rate (FAR),





**Figure 11.** Comparison of accumulated discharges ($\sum Q_{\mathrm{1D-2D}}$) at rainwater inlets during the 20220711 storm where positive values indicate inflows. (a)-(b). FSN simulated discharges. (c-f) Simulated discharge differences ($\sum \Delta Q_{\mathrm{1D-2D}}$) between FSN and FSR defined by FSN minus FSR. Basemaps are provided by Tianditu.

Critical Success Index (CSI) and Bias (BIAS) are selected to evaluate the model performance in inundation area simulations
(McGrath et al. (2018)), using results from the FSR approach as the reference:

$$\mathrm{POD} = \frac{\mathrm{TP}}{\mathrm{TP} + \mathrm{FN}} \tag{25}$$





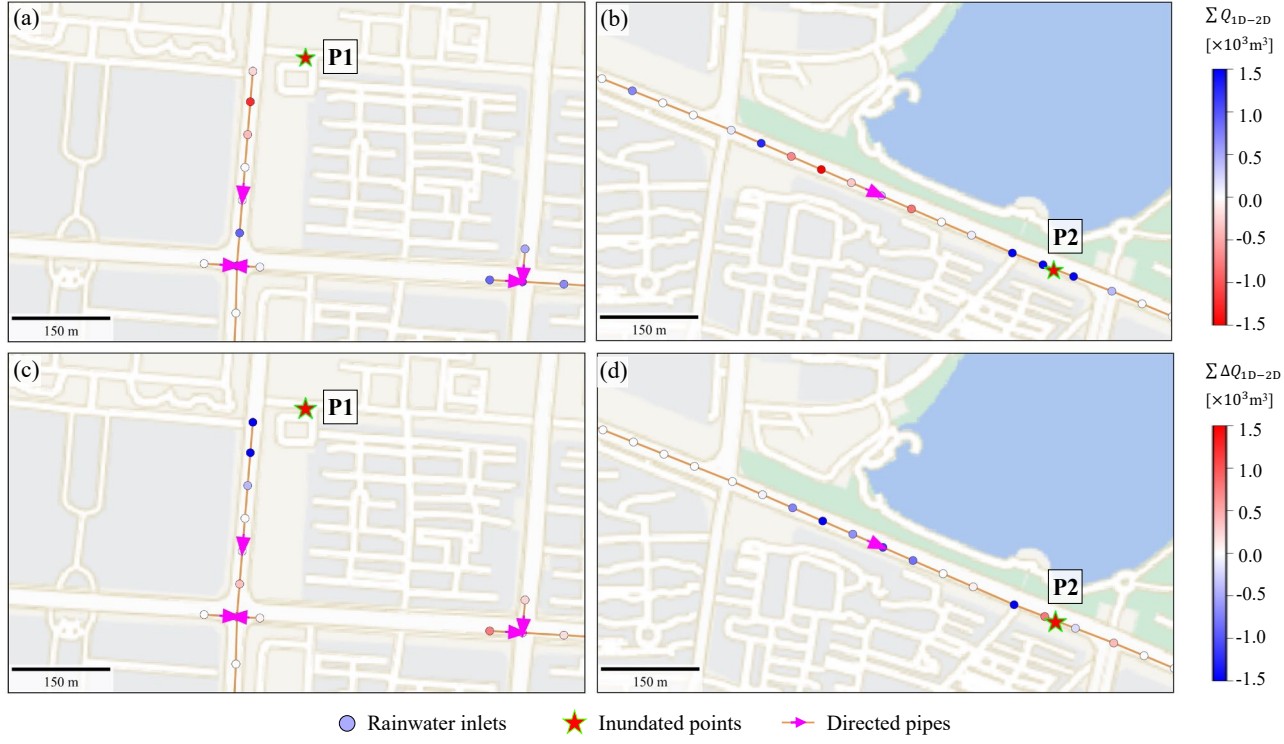

**Figure 12.** Comparison of accumulated discharges at rainwater inlets during the 20220711 storm. (a)-(b) RSN simulated discharges. (c)-(d) Simulated discharge differences between RSN and FSN defined by RSN minus FSN. Basemaps are provided by Tianditu.

$$\mathrm{FAR} = \frac{\mathrm{FP}}{\mathrm{TP} + \mathrm{FP}} \tag{26}$$

$$\mathrm{CSI} = \frac{\mathrm{TP}}{\mathrm{TP} + \mathrm{FN} + \mathrm{FP}} \tag{27}$$

$$\mathrm{BIAS} = \frac{\mathrm{TP} + \mathrm{FP}}{\mathrm{TP} + \mathrm{FN}} \tag{28}$$

where TP (True Positive) represents areas correctly identified as inundated, FN (False Negative) represents areas incorrectly classified as not inundated, and FP (False Positive) represents areas incorrectly classified as inundated, with FSR used as the reference. An area is considered inundated when its maximum depth exceeds 15 cm per local regulations.

In terms of detecting potential inundated areas, all approaches show only minor variations with high POD values ranging between 0.85 and 0.95. The NDA and RDA approaches even slightly outperform FSN and RSN under extreme rainfall con-

ditions, aligning with previous results in Sect. 4.3. As rainfall return periods decrease, FAR values of the RSN, NDA and





**Figure 13.** Comparison of simulated inundation area under design rainfall scenarios using different approximation methods.

RDA approaches gradually increase, with the latter two showing much more significant variations. This indicates that the performance of the equivalent drainage rate $i_d$ calibrated by a single event still has a high uncertainty when applied to varying rainfall scenarios. Therefore, while the difference in PODs are minimal, dramatically higher FARs of equifinal approaches widen the overall performance gap, as measured by CSI, especially for rainfall events with return periods below 50 years.

Regarding BIAS, physical approaches that adopt virtual networks but without controls consistently underestimate the total inundated area, leading to BIAS values less than 1. In contrast, other equifinal approaches tend to overestimate the inundated area. Specifically, RDA and NDA can overestimate the total inundated area by more than 8% and 10% for rainfall events with return periods less than 100 years, respectively. Thus, a constant equivalent drainage rate is insufficient to achieve an accu-





rate simulation of sewer drainage effects under diverse rainfall conditions, while physical approaches can also yield biased
estimation of the inundated area if regulated facilities are not properly configured.

From the perspective of water balance, total inundated areas (or volumes) are closely related to the volume of rainwater
drained outside the system. Thus, we further compare the simulated discharge processes at outflow points using different
physical approaches (Fig. 14). Being the solution reconstructed from the most detailed information, FSR clearly demonstrates
the insufficient drainage capacities at most outflow points such as OF1. When regulated facilities are excluded but sewer layout
remains unchanged, FSN yields significantly higher discharges, especially at major outflow points such as OF2 and OF3 with
peak flows over 300% higher, leading to an underestimation of total inundated areas. When sewer layout is further simplified
through major roads, the simulated outflows generally decrease by 27% due to less rainwater entering the sewer system.
However, certain outflow points, such as OF2, still show substantially increased discharges. Comparing the layouts used in
FSN and RSN reveals that a major pipe, measuring 6.8 m × 2.6 m (near OF1 and OF8, cf. Fig. 15) is found to be removed
during simplification, blocking the drainage pathway that divides the corresponding upstream rainwater to the northern outflow
points such as OF1 and OF8, eventually magnifying the inflow to OF3. Thus, for real urban environments with complicated
sewer layouts due to practical factors such as multistage construction, direct geometric simplification through major roads may
remove some critical connectivity inside a graph-like drainage system and thus fail to accurately represent the drainage effects
of the sewer network due to topological incompleteness.

## 5 Conclusions

In this work, we present a graph reconstruction-based approach for building a physical sewer hydraulic model, leveraging
incomplete information and surface elevation model while adhering to design guidelines. In contrast to conventional optimal-
design-based approaches, our approach seeks to reconstruct the most possible sewer network with related hydraulic properties,
such as gravitional flow direction and nodal invert elevation, constrained by minimal deviation from the initial status based on
available information. The resulting mathematical programming problems can be equivalently transformed into corresponding
linearized formulations, ensuring their computational efficiency for large-scale applications. We apply the proposed approach
to Yinchuan's main city zone, characterized by multiple outfalls consisting of wastewater treatment plants and pump stations,
yielding the following key insights:

- The reconstructed sewer hydraulic module integrated into a 1D/2D coupled hydrologic-hydrodynamic model achieves
high accuracy in simulating maximum inundation depths, with an $R^2$ of 0.95. A comparison with road-based layout
     simplifications further demonstrates the robustness of our approach in reconstructing sewer hydraulic properties, as the
     simplified model maintains a similar distribution of pipe slopes.

- While equifinal approaches ease parameter calibration by encapsulating complex sewer drainage processes into equiva-
     lent drainage rates, the proposed reconstruction approach can reliably achieve superior performance in simulating inun-
dated areas across various rainfall scenarios. This improvement stems from its physically grounded structure in simulat-



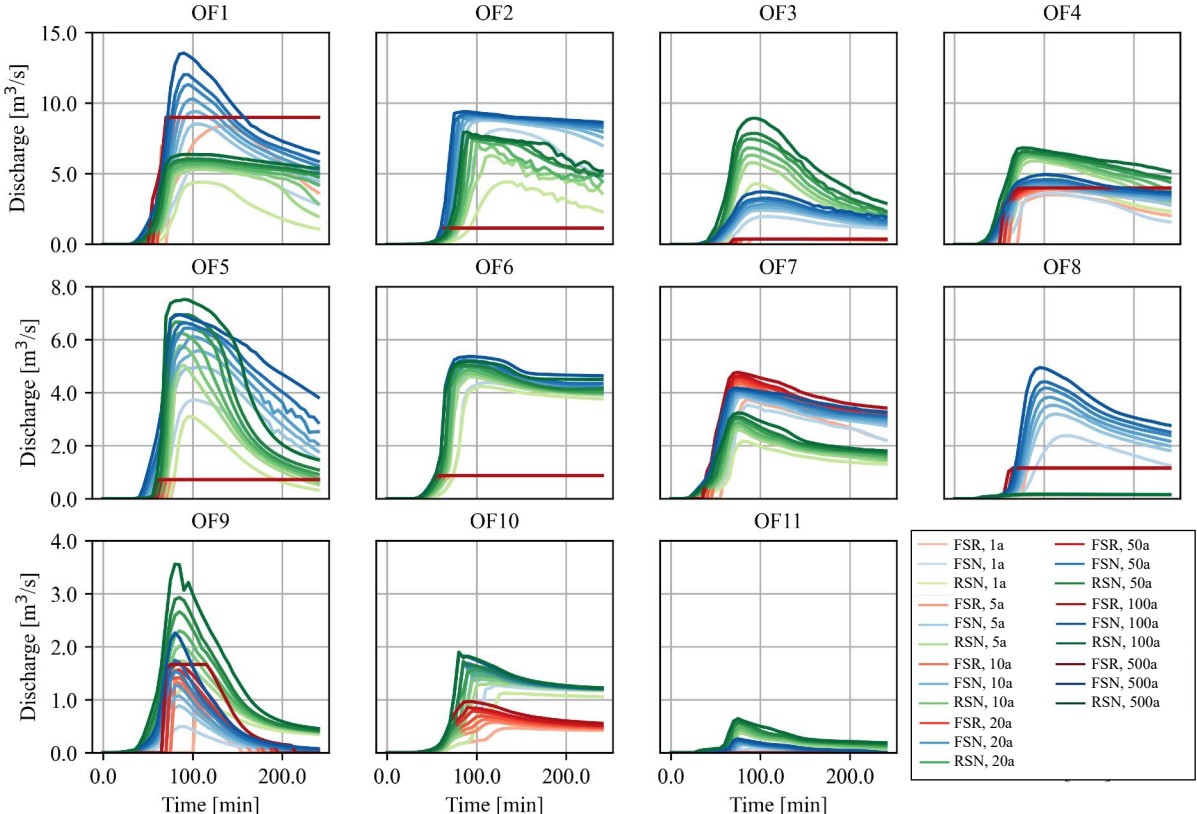

**Figure 14.** Comparison of simulated outflow processes under design rainfall scenarios using different approximation methods.

ing bidirectional interaction between the ground surface and the sewer system, indicating that physical reconstruction is preferable to approximate sewer drainage effects in operational applications.

– The completeness of available information has non-negligible effects on the simulated hydraulic performance of recon-
structed sewer networks. Simplifications involving road-based layouts and regulation facility removal lead to complex
deviation patterns in local maximum inundation depths and outflow hydrographs, with both overestimations and underes-
timations arising from the combined effects of surface microtopography, sewer connectivity modification and regulation
facility operation.

In conclusion, we provide an efficient sewer reconstruction approach that improves the simulation and understanding of sewer drainage effects during urban pluvial floods, accounting for varying levels of information completeness. However, un-
certainties persist in reconstructed physical models when detailed information on regulated facilities and drainage pathways is lacking. Thus, future research will focus on integrating advanced techniques, such as data assimilation, with reconstructed physical models to augment the realism of sewer drainage process representation.



*Code and data availability.* Landuse (https://worldcover2021.esa.int/download). DEM and sewer network information are not publicly available due to the privacy reasons. The code for end-to-end sewer reconstruction will be open once the manuscript is accepted.

**Appendix A: Slope estimation using multiple points**

Given a path consisting of $K$ points $p = \{v_1, ..., v_K\}$ with corresponding elevation values $z = \{z_1, ..., z_K\}$, its slope can be derived from the estimated coefficient of ordinary least squares regression through the coordinates $(v_r, z_r)$:

$$i_p = \frac{\sum_{r=1}^{K}(d_r - \mu_d)(z_i - \mu_z)}{\sum_{r=1}^{K}(d_r - \mu_d)^2} = \frac{\sum_{r=1}^{K} d_r z_r - K\mu_d\mu_z}{\sum_{i=1}^{K} d_r^2 - K\mu_d^2} \tag{A1}$$

where $d_r$ is the distance from $v_1$ to $v_r$ along the path ($d_1 = 0$), $\mu_d$ and $\mu_z$ is the mean value of $\{d_r\}_{r=1,..,K}$ and $\{z_r\}_{r=1,..,K}$.

Eq. A1 can re-written by using $z_r$ as pivot variables:

$$\begin{aligned}
i_p &= \frac{\sum_{r=1}^{K} d_r z_r - K\mu_d\mu_z}{\sum_{i=1}^{K} d_r^2 - K\mu_d^2} \\
&= \frac{K\sum_{r=1}^{K} d_r z_r - (\sum_{r=1}^{K} d_r)\cdot(\sum_{r=1}^{K} z_r)}{K\sum_{r=1}^{K} d_r^2 - (\sum_{r=1}^{K} d_r)^2} \\
&= \sum_{k=1}^{K} \frac{\left(K d_k - \sum_{r=1}^{K} d_r\right)}{K\sum_{r=1}^{K} d_r^2 - \left(\sum_{r=1}^{K} d_r\right)^2} z_k
\end{aligned} \tag{A2}$$

Thus, $i_p$ can be expressed as a linear combination of $z_r$. When $K = 2$, i.e, only two points are involved in the slope estimation, we can simplify Eq. A2 into Eq. A3:

$$i_p = -\frac{1}{d_2} z_1 + \frac{1}{d_2} z_2 \tag{A3}$$

*Author contributions.* RL conceived the idea and implement the algorithm. JL and JS collected the data. RL, TS, JL, FT, and GN conducted the analysis. RL drafted the manuscript and all authors reviewed and edited the manuscript.

*Competing interests.* At least one of the (co-)authors is a member of the editorial board of the Hydrology and Earth System Sciences.

*Acknowledgements.* This work was supported by the Gansu Province Science and Technology Department (22ZD6WA043), Program of Joint Research Institute of Tsinghua University—Ningxia Yinchuan for the Internet of Water and Digital Governance (SKL-IOW-2022TC2010),
NERC Independent Research Fellowship (NE/P018637/2).





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







**Figure 15.** Comparison of simulated flows at 90 percentiles ($Q_{90}$) of pipes under design rainfall scenarios using physical approaches with-/without network simplification. (a) FSN approach. (b) RSN approach. Basemaps are provided by Tianditu.