# Peer review of "Enhancing Urban Pluvial Flood Modelling through Graph Reconstruction of Incomplete Sewer Networks"

_EGUsphere, 2024_

## Author Response (AR1)

**Response to reviews on manuscript egusphere-2024-3780**

We appreciate the insightful comments from the editor and reviewers that have remarkably improved the quality of our manuscript. Please find below:

- our point-to-point responses (Sans Serif font in blue) to reviewer comments (RCs); and
- excerpts of revisions in salmon with a grey background, where necessary.

**Response to the Reviewers**

**Reviewer 1**

The manuscript is relevant as it aligns with the research on sewer network data scarcity. The authors propose a methodology for sewer network reconstruction using a graph-based approach. The results appear comprehensive and more accurate than other methodologies, such as generating a virtual network based on road layouts, as proposed in previous studies. Some minor comments could help improve the manuscript:

**Reply**: We appreciate your recognition of our work!

**RC 1.1** In Sect. 2.1, the authors introduce the definition of "source" (Line 93) but never reuse it again in the following analysis. Additionally, they also mention terms related to "inlet" like "Non-inlet nodes" (Line 118). Do they share the same meaning? If so, please revise the related sentences for consistency between definition and analysis.

**Reply**: Thanks for pointing out this typo. The concept of "source" is different from that of "inlet" where the former is defined by the in-degree and out-degree of nodes and the latter is associated with a specific semantic of the sewer system. For the topological constraint used in the gravitational flow direction derivation (Line 118 in the original manuscript), it should use the term of "source" and we have corrected it as follows (Line 125):

- Eq. 8e: Non-source and non-island nodes must have positive in-degrees (Eq. 4)

RC 1.2 For computational efficiency, the authors only use rough words like "a few minutes" without the information of specific time statistics and device specification (Line 292). If possible, please add more related details to offer readers/users clearer insights into the efficiency of algorithm execution.

**Reply**: Thank you for your concern about the computational configuration and related efficiency of our algorithm. We have clarified the time statistics and device specification as follows (Line 266-268):

Owing to linearized programming formulation used in the process of gravitational flow directions and nodal invert elevation reconstruction (Sect. 2.4), 1D sewer hydraulic models can be built from prepared geographical datasets in 33 seconds on a local desktop using a single core of Intel(R) Core(TM) i7-14700KF CPU and 400 MB RAM occupancy.

RC 1.3 In Sect. 4.3, the comparison results show that regulation facilities has significant effects on the results of inundation simulation. Thus, compared to the current comparison between FSN and RSN, it might be more valuable to consider the possible effects of road-base layout simplification with controls, given that the location of regulation facilities can be easily identified by field investigation and then incorporated into the flood model.

**Reply**: Thanks for your suggestions. We have added the baseline method that uses road-based sewer reconstruction with regulation facilities (denoted as RSC) in Sect. 3.4 as follows:

Five baselines are selected for the comparison of the performance between the proposed approach (denoted as the FSR approach) and other approximation methods:

- Full-Sewer-reconstruction-No-controls (FSN) which uses the same reconstruction approach but ignores the existence of regulated facilities such as pump stations and wastewater treatment plants. To be specific, regulated facilities located at outflow and intermediate points are configured as free outfalls and uncontrolled junctions, respectively.
- Road-based-Sewer-reconstruction-No-controls (RSN) which uses the same reconstruction approach but adopts the layout of sewers filtered by the major roadway zone (Fig. 4(b)) and ignores the existence of regulated facilities.
- Road-based-Sewer-reconstruction-with-Controls (RSC) which uses the reconstructed sewer network from RSN and further incorporates regulated facilities at corresponding locations.
- Road-Drainage Approximation (RDA) (Li et al., 2023; Xing et al., 2021) which subtracts the mass at a uniform rate  $i_d$  from the roadway zone buffered from road centerlines using widths evaluated by The Design of Urban Road Engineering (CJJ 37—2012).
- No-Drainage Approximation (NDA) which ignores the drainage effects of the sewer network.

According to simulation results, the RSC approach can lead to similar but slighter overestimation on the maximum inundation depth at points such as P2 (Sect. 4.3, Line 321-323):

Similar phenomenon of overestimation can also be observed when comparing RSN with RSC whereas with greatly decreased magnitude since road-based simplification reduces the rainwater collected into the sewer system and thus alleviates the drainage burden.

In addition, the RSC approach also demonstrates less influence of regulation facilities due to reduced rainwater flowing into the sewer system (Sect. 4.4, Line 375-379):

When the sewer layout is further simplified by the major roads, the simulated outflows decrease by approximately 27% due to reduced inlets that collect rainwater into the sewer system. This attribution can be further supported by the improved alignment of outflow hydrographs between RSN and RSC, which exhibit fewer truncations during peak flows under regulated conditions when compared to FSR and FSN. This indicates an alleviation of drainage deficiency due to reduced rainwater inflow.

RC 1.4 For Line 144, there might to be an extra ")"

**Reply**: Thanks for your advice on paper writing. We have corrected the sentences as follows (Line 150-152):

The objective functions of Eq. 8 and Eq. 9 are expressed as the sum of absolute deviations and can be further reduced into an equivalent linear form as follows (Giangrande et al., 2013; Wagner, 1959), thus ensuring their solution efficiency for large-scale applications.

**Reviewer 2**

The manuscript is relevant as it aligns with the research on sewer network data scarcity. The authors propose a methodology for sewer network reconstruction using a graph-based approach. The results appear comprehensive and more accurate than other methodologies, such as generating a virtual network based on road layouts, as proposed in previous studies. Some minor comments could help improve the manuscript:

**Reply**: We appreciate your recognition of our work!

RC 2.1 In Section 3.2, the manuscript explains how water is drained from the TSU to the network, but it does not clarify whether flow is generated from the network to the TSU when manholes are overloaded.

**Reply**: Thanks for pointing out this unclarified issue during surface-sewer flow coupling. We have clarified the bidirectional coupling scheme used by AUTOSHED in the Appendix B as follows (Line 452-458):

In AUTOSHED, the surface flow and sewer flow are bidirectionally linked via the mass source term  $Q_{1\mathrm{D-2D}}$  which can be parameterized as follows (Buttinger-Kreuzhuber et al., 2022):

$$Q_{\text{1D-2D}} = \begin{cases} \frac{2}{3} c_w P \sqrt{2g} h_{\text{2D}}^{1.5}, & H_{1D} < z_{\text{bc}} \\ c_{o,s} A \sqrt{2g} [h_{\text{2D}} - (H_{\text{1D}} - z_{\text{bc}})]^{0.5} & z_{\text{bc}} \le H_{\text{1D}} < z_{\text{bc}} + h_{\text{2D}} - A/P \\ c_{w,s} P \sqrt{2g} h_{\text{2D}} [h_{\text{2D}} - (H_{\text{1D}} - z_{\text{bc}})]^{0.5} & z_{\text{bc}} + h_{\text{2D}} - A/P \le H_{\text{1D}} < z_{\text{bc}} + h_{\text{2D}} \\ - c_{w,s} P \sqrt{2g} h_{\text{2D}} [(H_{\text{1D}} - z_{\text{bc}}) - h_{\text{2D}}]^{0.5} & z_{\text{bc}} + h_{\text{2D}} \le H_{\text{1D}} < z_{\text{bc}} + h_{\text{2D}} + A/P \\ - c_{o,s} A \sqrt{2g} [(H_{\text{1D}} - z_{\text{bc}}) - h_{\text{2D}}]^{0.5} & H_{\text{1D}} \ge z_{\text{bc}} + h_{\text{2D}} + A/P \end{cases}$$
(B1)

Here we assume  $Q_{\rm 1D-2D}$  is positive when water flows from the TSU to the linked inlet.  $H_{\rm 1D}$ ,  $h_{\rm 2D}$  and  $z_{\rm bc}$  are the total hydraulic head at the inlet, the water depth at the TSU and the center bed elevation at the TSU, respectively; A and A are the inlet's area and perimeter, respectively; A and A are discharge coefficients for the free weir, the submerged weir, and the orifice equations, respectively.

Thus, when  $H_{\rm 1D} < z_{\rm bc} + h_{\rm 2D}$ , we have positive  $Q_{\rm 1D-2D}$  for water flowing from the TSU to the manhole; when  $H_{\rm 1D} \geq z_{\rm bc} + h_{\rm 2D}$ , we have negative  $Q_{\rm 1D-2D}$  for water flowing from the manhole to the TSU.

RC 2.2 It is also recommended to explain whether the 2D-1D coupled model maintains mass balance.

**Reply**: Thank you for raising the issue of mass balance. To clarify why the 2D-1D coupled model can maintain mass balance, We have revised related sentences with typos and explained the reason in the Appendix B (Line 461-478):

After examining the links between TSUs and inlets, we find that each TSU is linked with at most one inlet and thus have:

$$q_{\rm sd} = \frac{Q_{\rm 1D-2D}}{A_{\rm T}} \tag{B2}$$

where  $A_{\rm T}$  is the total area of the TSU, and  $Q_{\rm 1D-2D}$  is the sewer drainage rate at the inlet derived from Eq. B1 and further constrained by the water availability of both the TSU and the inlet (Buttinger-Kreuzhuber et al., 2022):

$$Q_{1D-2D} = \begin{cases} \min\left(\frac{h_{2D}A_T}{\Delta t}, Q_{1D-2D}^*\right), & H_{1D} < z_{bc} + h_{2D} \\ \max\left(-\frac{V_{as}}{\Delta t}, Q_{1D-2D}^*\right), & H_{1D} \ge z_{bc} + h_{2D} \end{cases}$$
(B3)

where  $V_{\rm as}$  is the water volume at the inlet assembly determined by  $A_{\rm as}$  and  $H_{\rm 1D}$ ;  $\Delta t$  is the coupling time step evaluated as the minimum value between the timesteps of the 1D and 2D solvers (Chen et al., 2018).

Notably, Eq. B2 and Eq. B3 can ensure strict mass balance during surface-sewer flow coupling since it is straightforward to prove:

• The total exchange mass is the same between the surface and sewer network

$$q_{\rm sd}A_{\rm T}\Delta t = Q_{\rm 1D-2D}\Delta t \tag{B4}$$

• The total exchange mass does not exceed the available water in the surface when water flows from the TSU to the inlet, i.e.,  $H_{1D} < z_{bc} + h_{2D}$ ,

$$Q_{1D-2D}\Delta t = \min(h_{2D}A_T, Q_{1D-2D}^*) \le h_{2D}A_T$$
 (B5)

• The total exchange mass does not exceed the available water in the sewer network when water flows from the inlet to the TSU, i.e.,  $H_{\rm 1D} \geq z_{\rm bc} + h_{\rm 2D}$ ,

$$Q_{1D-2D}\Delta t = \max\left(-V_{as}, Q_{1D-2D}^*\right) \ge -V_{as}$$
(B6)

RC 2.3 In line 292, it is recommended to describe the specifications of the computer used for the analysis.

**Reply**: Thank you for your concern about the computational specifications of our algorithm. We have clarified the computational specifications as follows (line 266-268):

Owing to linearized programming formulation used in the process of gravitational flow directions and nodal invert elevation reconstruction (Sect. 2.4), 1D sewer hydraulic models can be built from prepared geographical datasets in 33 seconds on a local desktop using a single core of Intel(R) Core(TM) i7-14700KF CPU and 400 MB RAM occupancy.

RC 2.4 Clarify whether the methodology can be applied without information on pumps and WTPs. What alternatives exist if data on their location or technical specifications are unavailable?

**Reply**: Thanks for your advice on the applicability of our method when the information on regulation facilities is unavailable. We have clarified the necessary input of our method in Sect. 2.1 (Line 77-78):

The necessary inputs of sewer network reconstruction include the sewer network layout with specified outfalls and corresponding digital elevation model (DEM).

Thus, our method can be applied without information on pumps and WTPs. However, considering the non-negligible influence of regulation facilities such as pumps and WTPs on inundation simulation results, we also discuss some methods for related information collection in Sect. 4.5 as follows (Line 388-394):

The proposed graph reconstruction-based approach offers an efficient solution for generating physical sewer models using available data on sewer layouts and surface elevation, which can be obtained from local surveys (Xing et al., 2022), engineering drawings (Lyu et al., 2018), or open source software such as SWMManywhere (Dobson et al., 2025). Given the significant influence of network topology and regulation facilities on inundation simulation results (as discussed in Sect. 4.3 and 4.4), in addition to the aforementioned data sources, it is imperative to further implement intelligent surveying systems targeting relevant critical components such as inlets and outfalls. These data can then be integrated with the proposed reconstruction algorithm to achieve more reliable pluvial flood modeling.

**Reviewer 3**

This paper presents a graph reconstruction-based approach for generating physically realistic sewer models from incomplete information, for addressing a significant challenge in urban flood modeling. The authors develop a methodology that leverages graph theory and linearized programming formulations to derive gravitational flow directions and nodal invert elevations in decentralized sewer networks with multiple outfalls. They conduct a case study in Yinchuan, China, by performing a 1D/2D coupled hydrologic-hydrodynamic model and reproduces maximum inundation depths with high accuracy ( $R^2 = 0.95$ ) when complete network information is available. The paper's strengths lie in its mathematical formulation that transforms complex nonlinear problems into computationally efficient linear equivalents, and its comprehensive comparison against alternative approaches across varying rainfall scenarios. However, there are some major issues that authors need to address before publication.

**Reply**: We appreciate your recognition of our work!

RC 3.1 The paper lacks technical discussion on the implementation, i.e., the solver used for the linearized formulations. There are several existing solvers, each with their own strengths, weaknesses, and parameters requiring tuning. Sometimes solvers require citation too, so the authors need to credit the developers of packages and software used to develop their tool.

**Reply**: Thanks for pointing out this unclarified issue. We have clarified the solver used for linear programming in Sect. 2.4 (Line 163-165)

In terms of solving the reformulated linear programming problems, we select the open-source Computational Infrastructure for Operations Research (COIN-OR) branch-and-cut solver (CBC) (Forrest and Lougee-Heimer, 2005) through its Python interface in PuLP (Mitchell et al., 2011).

RC 3.2 While authors discuss briefly that their approach is fast, they do not provide any supporting evidence, especially since the source code is not available for review. The authors mention they will open source their code, but it's common practice nowadays for authors to provide access to the code that was used to generate the results during the review process for reproducibility and validation.

**Reply**: Thank you for raising the issue of reproducibility and validation. We have clarified the time statistics with corresponding device specification as follows (Line 266-268):

Owing to linearized programming formulation used in the process of gravitational flow directions and nodal invert elevation reconstruction (Sect. 2.4), 1D sewer hydraulic models can be built from prepared geographical datasets in 33 seconds on a local desktop using a single core of Intel(R) Core(TM) i7-14700KF CPU and 400 MB RAM occupancy.

Furthermore, we have published the related code for reproducibility and validation as follows (Line 438-440):

The snapshot of the source code and compiled application for sewer reconstruction has been archived on Zenodo (https://doi.org/10.5281/zenodo.15522608). And the up-to-date version is available at: https://github.com/L11C-mmd/ASHED\_sewerReconstruction.

RC 3.3 The authors do not provide any discussion on the sampling approach used for getting node elevations from the DEM. DEMs, especially in urban areas, can contain artifacts that if not addressed through DEM processing operations such as conditioning can lead to propagation of uncertainties in raw DEMs into the generated network. While the algorithm provided by the authors tries to address the hydraulic feasibility of the network, if the source DEM contains major artifacts, especially for high-resolution DEMs in built areas, it can adversely impact the quality of the generated network.

**Reply**: Thanks for your concerns on nodal elevation derivation from DEM. As stated in Sect. 2.1 (Line 105), we get the elevation of the nodes by matching their coordinates with the pixels of the DEM. And we have added the necessary introduction to the related DEM preprocessing procedures in Sect. 3.3 as follows (Line 208-211):

Following common practices of DEM preprocessing in urban flood simulation, we delete the inaccurately measured building roof elevation from the original product (Chen and Huang, 2024) and lower the surface elevation of grids containing roads by 0.2 m (Liu et al., 2022) to improve the representation of hydrological connectivity in DEM.

According to the preprocessed DEM, we can achieve a reasonable reconstruction of the sewer network where all proposed topological and slope constraints are satisfied. Furthermore, we have also acknowledged the potential limitation of input data quality in Section 4.5 (Line 397-406).

In addition, it should be acknowledged that considering inherent quality issues of multi-source geographical datasets, mathematical programming procedures in the proposed approach may occasionally fail because of infeasible configurations such as over-elevated outfalls and dangling pipelines. While introducing data cleaning procedures may help alleviate such issues and streamline the downstreaming reconstruction process, it also necessitates the trade-off between algorithm robustness and accuracy owing to embedded assumptions during quality control, which deserves further investigations. In addition to related cleaning functions implemented in existing packages like SWMManywhere (Dobson et al., 2025), we propose leveraging the built-in functionality of irreducible inconsistent subset (IIS) computation within our optimization-based framework, which can be steadily accessed in solvers such as Gurobi (https://docs.gurobi.com/projects/optimizer). This tool can help identify the specific constraints responsible for infeasibility and thus support the spatial pinpointing of ill-conditioned sewer elements that over-constrain the optimization and may require further correction or elimination.

By IIS computation, we can efficiently identify the nodes associated with potentially problematic elevations and perform targeted corrections. We have offered one case (https://github.com/LllC-mmd/ASHED\_sewerReconstruction/tree/main/example/case0) in our published codebase to exemplify related usages.

RC 3.4 Pages 9-11 are allocated to providing mathematical details of the AUTOSHED model. It's not clear from the paper whether the authors actually made any developments in the coupled 1D/2D model, or they are just simply using the model as an off-the-shelf tool. If they are simply using the model, there's really no need for this detailed discussion, and can even be misleading, as the authors can mention some relevant high-level information since the focus of this study seems to be on generating a hydraulically feasible network.

**Reply**: Thanks for your advice on the paper writing. We have simplified the description of coupled 1D/2D flood simulation by keeping necessary governing equations in the main sections (Sect. 3.1 and 3.2) and moving related solution details into Appendix B.

RC 3.5 The discussion on the outfall is not very clear. Do the authors assume that the outfall locations are known? If that's the case, this can be a major limitation as they are often not readily available. If the authors have some way of identifying the outfall locations, they must explicitly discuss the details; otherwise, this should be explicitly mentioned as one of the limitations as the authors state that one of their major contributions is deriving hydraulically feasible decentralized sewer networks with multiple outfalls.

**Reply**: Thanks for your concerns on the outfall identification. We have clarified the necessary input of our method in Sect. 2.1 (Line 77-78):

The necessary inputs of sewer network reconstruction include the sewer network layout with specified outfalls and corresponding digital elevation model (DEM).

For our study area, we collect these input datasets from field investigation and design documents (Line 217-218):

Sewer network layout and corresponding attribute of pipeline sizes (Fig. 4 (a)) digitized from investigation and design documents authorized by the local government.

Furthermore, we also have added the related discussion of outfall identification in Sect. 4.5 as follows (Line 388-397):

The proposed graph reconstruction-based approach offers an efficient solution for generating physical sewer models using available data on sewer layouts and surface elevation, which can be obtained from local surveys (Xing et al., 2022), engineering drawings (Lyu et al., 2018), or open source software such as SWMManywhere (Dobson et al., 2025). Given the significant influence of network topology and regulation facilities on inundation simulation results (as discussed in Sect. 4.3 and 4.4), in addition to the aforementioned data sources, it is imperative to further implement intelligent surveying systems targeting relevant critical components such as inlets and outfalls. These data can then be integrated with the proposed reconstruction algorithm to achieve more reliable pluvial flood modeling. In particular, outfall identification still remains a difficult task, especially for decentralized sewer networks in relatively flat urban areas, challenging the applicability of existing topography-driven approaches (Dobson et al., 2025) and motivating the need for user-defined configurations according to prior knowledge of the area (Reyes-Silva et al., 2023).

RC 3.6 It appears the algorithm has only one tunable parameter: pipe segmentation, with a default value of 50 m. However, the authors did not provide a discussion or perform a formal sensitivity analysis on this parameter. It's important to quantify how this parameter impacts the model performance.

**Reply**: Thanks for your concerns about the sensitivity analysis of the pipe segmentation parameter. We have added the discussion about its influence on model simulation results in Sect. 4.2 (Line 290-296) as follows:

According to Fig. 9 (a), the FSR approach achieves an R2 of 0.95, which indicates good agreement between the simulated and observed values and thus consolidates the physical background for further comparison. In addition, we further examine the performance of the FSR approach with varying inlet spacings (denoted as ILS) from 30 m to 100 m. According to Fig. 9 (b), the FSR approach demonstrates relatively robust performance with R2 ranging from 0.85 to 0.95 where ILS of 50 m achieves the highest NSE. It is worth noting that increasing ILS leads to both underestimation and overestimation, such as P5 and P2 in Fig. 9 (b), highlighting the dual role of rainwater inlets in either alleviating surface inundation through drainage or contributing to it when overloaded.

RC 3.7 The authors do not provide a direct validation of their approach. They do so indirectly by measuring simulation results with inundation depths. While I understand that there aren't many public sewer data with actual flow directions, but since the contribution of authors is mainly related to enforcing correct flow directions throughout the network, and they perform pipe-level simulations, lack of validation with existing pipe flow directions is one of the major limitations that needs to be explicitly mentioned. If authors have access to such data, even partially, direct validation of flow direction can be helpful for quantifying the performance.

**Reply**: Thanks for your concerns about the direct validation of flow direction. In our proposed approach, during deriving optimal gravitational flow directions, if there is any known flow direction, it can be explicitly enforced as the constraints for optimization (Eq. 8d in Sect 2.2):

$$\overrightarrow{e}_{i,j} = \overrightarrow{e}_{i,j}^*, \forall \overrightarrow{e}_{i,j} \in \mathcal{E}_D^*$$
(8d)

This ensures strict consistency between the existing pipe flow directions and the optimized results, thereby eliminating the need for any additional validation steps. However, we recognize that in some cases, existing flow direction data may conflict with the topological feasibility of the sewer network, potentially causing the failure of the optimization process due to over-constraining. In such situations, further correction of the known flow directions may be necessary. We have acknowledged this limitation of input data quality and suggest a corresponding solution based on the concept of irreducible inconsistent subsets in Section 4.5 (Line 397-406).

In addition, it should be acknowledged that considering inherent quality issues of multi-source geographical datasets, mathematical programming procedures in the proposed approach may occasionally fail because of infeasible configurations such as over-elevated outfalls and dangling pipelines. While introducing data cleaning procedures may help alleviate such issues and streamline the downstreaming reconstruction process, it also necessitates the trade-off between algorithm robustness and accuracy owing to embedded assumptions during quality control, which deserves further investigations. In addition to related cleaning functions implemented in existing packages like SWMManywhere (Dobson et al., 2025), we propose leveraging the built-in functionality of irreducible inconsistent subset (IIS) computation within our optimization-based framework, which can be steadily accessed in solvers such as Gurobi (https://docs.gurobi.com/projects/optimizer). This tool can help identify the specific constraints responsible for infeasibility and thus support the spatial pinpointing of ill-conditioned sewer elements that over-constrain the optimization and may require further correction or elimination.

RC 3.8 In the introduction, I suggest providing a discussion on how this work compares to more recent relevant work in the literature on this topic, such as https://doi.org/10.1016/j.envsoft.2025.106358.

**Reply**: Thanks for your advice. We have included the discussion on the differences between this work and SWMManywhere in the literature review as follows (Line 59-65):

Although existing tools such as SWMManywhere (Dobson et al., 2025) have offered practical solutions for high-quality VND using publicly available datasets, certain discrepancies between virtual and real networks still inevitably exist due to inherent assumptions and simplifications, such as predefined road alignments for network topology, river proximity for outfall location, and ideal hydraulic design using the Rational Method. These discrepancies can potentially alter hydrological connectivity and lead to undesirable performance in flood simulation (Tran et al., 2024). Thus, in order to enhance the physical realism of virtual networks, additional efforts should be devoted to incorporating partially known information as constraints of network generation.

RC 3.9 For all figures containing a basemap, make sure the attribution follows the specific language required by Esri.

**Reply**: Thanks for your suggestions on the basemap attribution. We have revised the attribution sentence following the specific language required by Esri. An example of Fig. 3 is given as follows:

Geographical datasets in the study area. (a) Digital elevation model. (b) Land use map and road networks. Basemaps are derived from ESRI World Imagery (Credit: Esri, TomTom, Garmin, FAO, NOAA, USGS, © OpenStreetMap contributors, and the GIS User Community).

RC 3.10 Throughout the text, some citations are not correct. For example, on line 144 the citation is "(Wagner (1959); Giangrande et al. (2013)" which not only misses a parenthesis but also is not in the correct format which is "(Wagner, 1959; Giangrande et al., 2013)". This should be a LaTeX-related issue. Please carefully go over the text to catch citations with similar issues, as I found many instances, e.g., on pages 10 and 11.

**Reply**: Thanks for your advice on the paper writing. We have corrected related sentences as follows (Line 150-152):

The objective functions of Eq. 8 and Eq. 9 are expressed as the sum of absolute deviations and can be further reduced into an equivalent linear form as follows (Giangrande et al., 2013; Wagner, 1959), thus ensuring their solution efficiency for large-scale applications.

We also have carefully gone over the text to eliminate similar issues.

RC 3.11 There are some awkward sentences throughout the text. Please carefully proofread the paper.

**Reply**: Thanks for your suggestions. We have carefully proofread the paper and improved the quality of the paper writing.

RC 3.12 What does the "equifinal method/approach" mean? Does it mean conceptual models? While equifinality is commonly used in the hydrology literature, I am not familiar with using equifinal method in the context of different modeling approaches. Please clarify this term or replace it with a more common jargon.

**Reply**: Thanks for your suggestions on the clarification of the "equifinal method/approach" term. We have clarified in the revised manuscript as follows (Line 244-247):

According to whether a sewer hydraulic module is reconstructed for simulation with AUTOSHED, we can further classify the above approximation methods into two categories: physical approaches (including FSR, FSN, RSN and RSC), and equifinal ones (including RDA and NDA), meaning similar flood simulation results can be achieved by different methods with appropriate parameters of drainage effects (Dobson et al., 2025).

RC 3.13 Fig 8: Use a different legend, it's very difficult to see what's going on.

**Reply**: Thanks for your suggestions on the figure plotting. We have updated the legend of inundated points with a more contrastive color as follows.

**References**

- Buttinger-Kreuzhuber, A., Konev, A., Horváth, Z., Cornel, D., Schwerdorf, I., Blöschl, G., and Waser, J.: An integrated GPU-accelerated modeling framework for high-resolution simulations of rural and urban flash floods, Environmental Modelling & Software, 156, 105 480, https://doi.org/https://doi.org/10.1016/j.envsoft.2022.105480, 2022.
- Chen, W., Huang, G., Zhang, H., and Wang, W.: Urban inundation response to rainstorm patterns with a coupled hydrodynamic model: A case study in Haidian Island, China, Journal of Hydrology, 564, 1022–1035, https://doi.org/https://doi.org/10.1016/j.jhydrol.2018.07.069, 2018.
- Chen, Z. and Huang, G.: Numerical simulation study on the effect of underground drainage pipe network in typical urban flood, Journal of Hydrology, 638, 131 481, https://doi.org/https://doi.org/10.1016/j.jhydrol.2024.131481, 2024.
- Dobson, B., Jovanovic, T., Alonso-Álvarez, D., and Chegini, T.: SWMManywhere: A workflow for generation and sensitivity analysis of synthetic urban drainage models, anywhere, Environmental Modelling & Software, 186, 106358, https://doi.org/https://doi.org/10.1016/j.envsoft.2025. 106358, 2025.
- Forrest, J. and Lougee-Heimer, R.: CBC User Guide, chap. Chapter 10, pp. 257–277, INFORMS, https://doi.org/https://doi.org/10.1287/educ.1053.0020, 2005.
- Giangrande, S. E., McGraw, R., and Lei, L.: An Application of Linear Programming to Polarimetric Radar Differential Phase Processing, Journal of Atmospheric and Oceanic Technology, 30, 1716 1729, https://doi.org/10.1175/JTECH-D-12-00147.1, 2013.

Figure 8: Maximum inundation depth map simulated by the FSR approach under the 20220711 storm. Basemaps are derived from ESRI World Imagery (Credit: Esri, TomTom, Garmin, FAO, NOAA, USGS, © OpenStreetMap contributors, and the GIS User Community).

- Li, D., Hou, J., Shen, R., Li, B., Tong, Y., and Wang, T.: Approximation method for the sewer drainage effect for urban flood modeling in areas without drainage-pipe data, Frontiers in Environmental Science, 11, https://doi.org/https://doi.org/10.3389/fenvs.2023.1134985, 2023.
- Liu, L., Sun, J., and Lin, B.: A large-scale waterlogging investigation in a megacity, Natural Hazards, 114, 1505–1524, https://doi.org/https://doi.org/10.1007/s11069-022-05435-3, 2022.
- Lyu, H., Ni, G., Cao, X., Ma, Y., and Tian, F.: Effect of Temporal Resolution of Rainfall on Simulation of Urban Flood Processes, Water, 10, https://doi.org/https://doi.org/10.3390/w10070880, 2018.
- Mitchell, S., OSullivan, M., and Dunning, I.: Pulp: a linear programming toolkit for python, The University of Auckland, Auckland, New Zealand, 65, 25, 2011.
- Reyes-Silva, J. D., Novoa, D., Helm, B., and Krebs, P.: An Evaluation Framework for Urban Pluvial Flooding Based on Open-Access Data, Water, 15, https://doi.org/https://doi.org/10.3390/w15010046, 2023.

- Tran, V. N., Ivanov, V. Y., Huang, W., Murphy, K., Daneshvar, F., Bednar, J. H., Alexander, G. A., Kim, J., and Wright, D. B.: Connectivity in urbanscapes can cause unintended flood impacts from stormwater systems, Nature Cities, 1, 654–664, https://doi.org/https://doi.org/10.1038/s44284-024-00116-7, 2024.
- Wagner, H. M.: Linear Programming Techniques for Regression Analysis, Journal of the American Statistical Association, 54, 206–212, https://doi.org/10.1080/01621459.1959.10501506, 1959.
- Xing, Y., Shao, D., Yang, Y., Ma, X., and Zhang, S.: Influence and interactions of input factors in urban flood inundation modeling: An examination with variance-based global sensitivity analysis, Journal of Hydrology, 600, 126 524, https://doi.org/https://doi.org/10.1016/j.jhydrol.2021. 126524, 2021.
- Xing, Y., Shao, D., Liang, Q., Chen, H., Ma, X., and Ullah, I.: Investigation of the drainage loss effects with a street view based drainage calculation method in hydrodynamic modelling of pluvial floods in urbanized area, Journal of Hydrology, 605, 127365, https://doi.org/https://doi.org/10.1016/j.jhydrol.2021.127365, 2022.

---

## Author Response (AR2)

**Response to reviews on manuscript egusphere-2024-3780**

We appreciate the insightful comments from the editor that have remarkably improved the quality of our manuscript. Please find below:

- our point-to-point responses (Sans Serif font in blue) to editor comments (ECs); and
- excerpts of revisions in salmon with a grey background, where necessary.

Editor Comment — The reviewers suggest accepting the paper in its current form, but I do have some suggestions and edit requests from my own reading that I would like the authors to consider before the paper can be accepted for publication in HESS.

Reply: We appreciate your recognition of our work!

**EC1** Section 2.1: This section is somewhat difficult to follow. I suggest expanding the explanation of the example given, and better aligning it with the text. Please also enhance the caption of Figure 1 with more descriptive information to help readers follow the formulation more easily.

**Reply**: Thank you for your suggestion. We have revised the caption of Figure 1 as follows:

Figure 1: Cycles, islands, outfalls and sources in the graph-based sewer network representation.

Furthermore, we have revised the introduction of graph representation of sewer networks with necessary explanation of Figure 1, e.g., Line 86-88:

Considering the potential cycles in the geometric layout of G, we identify the set of simple cycles  $\mathcal{C} = \{C_k\}$  defined as the set of closed paths where no node appears twice (Johnson, 1975), such as  $C_1$  and  $C_2$  in Fig. 1.

**Line 94-95:**

According to Eq. 3, we define a cycle  $C_k$  as an "island" if the sum of its in-degree and out-degree is equal to 1, such as  $C_2$  in Fig. 1.

**Line 98-99:**

For a node  $v_i \in V$ , we include it in the set of outfalls (denoted as  $\mathcal{O}$ ) if its out-degree is 0 and indegree is 1, such as  $v_{13}$  in Fig. 1, or in the set of sources (denoted as  $\mathcal{S}$ ) if its in-degree is 0 and out-degree is 1, such as  $v_1$  in Fig. 1

By expanding the explanation of the example given as above, we hope we can make this section easier to follow.

**EC2** The meaning of constraints 4 and 5 (Section 2.2) is not entirely clear. Please revise the explanation in the text for clarity. In addition, I recommend summarizing Equations 8a–8h in a table format, which could help the reader better understand the logic of the constraints.

**Reply**: Thank you for your suggestion. We have summarized Equations 8a-8h as Table 1 with rephrased descriptions of physical meanings of Equations 8f-8g as follows:

Table 1: Optimization constraints of gravitational flow direction derivation.  $|C_k|$  is the number of nodes in  $C_k$ ;  $\mathcal{E}_D^*$  is the set of links with known directions (denoted by  $\overrightarrow{e}_{i,j}^*$ );  $\overrightarrow{e}_{\text{CV}_k^{(|\mathcal{C}_k|)},\text{CV}_k^{(1)}} + \sum_{i=1}^{|\mathcal{C}_k|-1} \overrightarrow{e}_{\text{CV}_k^{(i)},\text{CV}_k^{(i+1)}}$  is the sum of consecutively connected direction values along  $C_k$ .

| Constraint                                                                                                                                                                                    | Description                                       |
|-----------------------------------------------------------------------------------------------------------------------------------------------------------------------------------------------|---------------------------------------------------|
| $\overrightarrow{e}_{i,j} = 1 - \overrightarrow{e}_{j,i}$                                                                                                                                     | No bi-directional links between consecutive nodes |
| $\overrightarrow{e}_{i,j} = 1, orall v_j \in \mathcal{O}$                                                                                                                                    | Outfalls must have zero out-degree                |
| $\overrightarrow{e}_{i,j} = 1, orall v_j \in \mathcal{O} \ \overrightarrow{e}_{i,j} = \overrightarrow{e}_{i,j}^*, orall \overrightarrow{e}_{i,j} \in \mathcal{E}_D^*$                       | Preserves known link directions                   |
|                                                                                                                                                                                               | Nodes except for sources and                      |
| $d_{\mathrm{in},i} \ge 1, \forall v_i \notin \mathcal{S} \cup \mathrm{ISD}(G)$                                                                                                                | those located in the islands must have            |
|                                                                                                                                                                                               | positive in-degrees (Eq. 4)                       |
| $d_{\mathrm{out},i} \ge 1, \forall v_i \notin \mathcal{O}$                                                                                                                                    | Nodes except for outfalls must have               |
|                                                                                                                                                                                               | positive out-degrees                              |
| $dC_{\mathrm{out},k} \ge 1, \forall C_k \in \mathcal{C}$                                                                                                                                      | Cycles must have positive out-degrees             |
|                                                                                                                                                                                               | for flow passage                                  |
| $\overrightarrow{e}_{\text{CV}_k^{( C_k )}, \text{CV}_k^{(1)}} + \sum_{i=1}^{ C_k -1} \overrightarrow{e}_{\text{CV}_k^{(i)}, \text{CV}_k^{(i+1)}} \le  C_k  - 1, \forall C_k \in \mathcal{C}$ | No recirculation within cycles                    |

**EC3** It would be helpful to include a schematic figure that illustrates a typical TSU and the fluxes computed by AUTOSHED, as described in Sections 3.1 and 3.2.

**Reply**: Thank you for your suggestion. We have added one schematic figure (Fig. 3) to illustrate the computation procedures within one TSU with related fluxes as follows:

EC4 Sections 3.3 and 3.4 describe the study area and baseline configurations, and are not directly related to the description of AUTOSHED in the earlier subsections. I suggest moving them to a new section (e.g., Section 4.1 and 4.2).

Figure 3. Flux calculation within a typical triangular-shaped unit and between surface-sewer system.

**Reply**: Thank you for your suggestion. We have moved Sect. 3.3 and 3.4 as Sect. 4.1 and 4.2.

**EC5** Figure 5 could be moved to the supporting material, as it is mostly illustrative and not essential for understanding the main text.

**Reply**: Thank you for your suggestion. We have moved Figure 5 as Figure S1 in the Supplementary Material and revised the related sentence.

**EC6** Figure 7 is currently not referenced in the main text. Additionally, it should be moved to the supporting material.

**Reply**: Thank you for your suggestion. This is due to one typo in Sect. 4.3. We have corrected the related reference to Figure 7 as follows (Line 251-252):

According to reconstructed flow directions, nodal invert elevation is further derived from values initialized by a constant cover depth (Eq. 6) and the slope distributions before / after adjustment are summarized in Fig. 7.

EC7 Rather than referring to the storm as "20220711", please consider using a more intuitive name, such as "the July 2022 storm" or simply "the case study storm".

**Reply**: Thank you for your suggestion. We have revised the related sentences as follows (Line 267-268, Line 278-280):

To facilitate the comparison of pluvial flood simulation performance using different approaches, we first calibrate the model parameters with a severe storm hitting CYC on 11 July 2022 (denoted as the July 2022 storm).

Fig. 8 shows the inundation map simulated by AUTOSHED with the sewer hydraulic module reconstructed from the FSR approach under the July 2022 storm.

**EC8** Table 2 can be moved to the supporting material, as it contains standard parameter values and is not central to the main findings.

**Reply**: Thank you for your suggestion. We have moved Table 2 as Table. S1 in the Supplementary Material.

**EC9** Figure 9 can be made smaller. The three subplots can be arranged in a single row. Aim for an overall size of approximately 8–10 cm height and 14–16 cm width.

**Reply**: Thank you for your suggestion. We have resized the figure into one row.

**EC10** Figure 10 should be placed in the supporting material.

**Reply**: Thank you for your suggestion. We have moved Figure 10 as Figure S2 in the Supplementary Material and revised the related sentence.

**EC11** Lines 346–356: The text in this paragraph is more appropriate for the Methods section and should be relocated accordingly.

**Reply**: Thank you for your suggestion. We have moved related sentences to Sect. 4.2 as follows (Line 239-242):

In order to quantify the simulation performance of different approximation methods under design scenarios, we select the Probability of Detection (POD), False Alarm Rate (FAR), Critical Success Index (CSI) and Bias (BIAS) of inundation area simulations (McGrath et al., 2018), using results from the FSR approach as the reference (see Sect. S3 in the Supplementary Material).

**EC12** Equations 18–21 are standard and do not necessarily need to appear in the main text. I suggest moving them to the supporting material.

**Reply**: Thank you for your suggestion. We have moved Equations 18-21 to Sect. S3 in the Supplementary Material.

**EC13** Figure 13: Consider resizing this figure to a smaller format. All four subplots can fit into one row, with a maximum height of 8–10 cm.

**Reply**: Thank you for your suggestion. We have resized the figure into one row.

**EC14** Figure 14 is detailed but not essential to the main discussion; it can be moved to the supporting material.

**Reply**: Thank you for your suggestion. We have moved Figure 14 as Figure S3 in the Supplementary Material and revised the related sentence as follows (Line 356-357):

Thus, we further compare the discharge processes at outfalls simulated by different physical approaches (see Fig. S3 in the Supplementary Material).

**References**

Johnson, D. B.: Finding All the Elementary Circuits of a Directed Graph, SIAM Journal on Computing, 4, 77–84, https://doi.org/10.1137/0204007, 1975.

McGrath, H., Bourgon, J.-F., Proulx-Bourque, J.-S., Nastev, M., and Abo El Ezz, A.: A comparison of simplified conceptual models for rapid web-based flood inundation mapping, Natural Hazards, 93, 905–920, https://doi.org/10.1007/s11069-018-3331-y, 2018.